# Effect of biannual azithromycin distribution on antibody responses to malaria, bacterial, and protozoan pathogens in Niger

Ahmed M. Arzika[1], Ramatou Maliki[1], E. Brook Goodhew[2], Eric Rogier[2], Jeffrey W. Priest [3], Elodie Lebas[4], Kieran S. O'Brien [4,5], Victoria Le [4], Catherine E. Oldenburg[4,5,6], Thuy Doan [4,5], Travis C. Porco[4,5,6], Jeremy D. Keenan [4,5], Thomas M. Lietman [4,5,6], Diana L. Martin[2,7], Benjamin F. Arnold [4,5,7 ✉] & MORDOR-Niger Study Group*

The MORDOR trial in Niger, Malawi, and Tanzania found that biannual mass distribution of azithromycin to children younger than 5 years led to a 13.5% reduction in all-cause mortality (NCT02048007). To help elucidate the mechanism for mortality reduction, we report IgG responses to 11 malaria, bacterial, and protozoan pathogens using a multiplex bead assay in pre-specified substudy of 30 communities in the rural Niger placebo-controlled trial over a three-year period ($n = 5642$ blood specimens, $n = 3814$ children ages 1–59 months). Mass azithromycin reduces *Campylobacter* spp. force of infection by 29% (hazard ratio = 0.71, 95% CI: 0.56, 0.89; $P = 0.004$) but serological measures show no significant differences between groups for other pathogens against a backdrop of high transmission. Results align with a recent microbiome study in the communities. Given significant sequelae of *Campylobacter* infection among preschool aged children, our results support an important mechanism through which biannual mass distribution of azithromycin likely reduces mortality in Niger.

---

[1] The Carter Center, Niger Niamey, Niger. [2] Division of Parasitic Diseases and Malaria, Centers for Disease Control and Prevention, Atlanta, GA, USA. [3] Division of Foodborne, Waterborne and Environmental Diseases, Centers for Disease Control and Prevention, Atlanta, GA, USA. [4] Francis I. Proctor Foundation, University of California, San Francisco, CA, USA. [5] Department of Ophthalmology, University of California, San Francisco, CA, USA. [6] Department of Epidemiology and Biostatistics, University of California, San Francisco, CA, USA. [7] These authors contributed equally: Diana L. Martin, Benjamin F. Arnold. *A list of authors and their affiliations appears at the end of the paper. ✉email: ben.arnold@ucsf.edu

Child mortality rates in Niger are among the highest in Africa[1]. We previously conducted a cluster-randomized, placebo-controlled trial in Niger, Malawi, and Tanzania to assess the effect of biannual mass distribution of azithromycin on mortality among preschool-aged children (MORDOR)[2]. In Niger, azithromycin reduced all-cause mortality by 18%, spurring interest in identifying specific mechanisms of pathogen reduction through which azithromycin reduced mortality. Follow-up analyses of MORDOR Niger demonstrated reductions in cause-specific mortality that spanned many leading causes of death in the region (e.g., malaria, pneumonia, and diarrhea), suggesting effects were unlikely to result from a single mechanism[3]. An intensive monitoring trial in MORDOR Niger also documented lower levels of malaria parasitemia[4] and reduced carriage of *Campylobacter upsaliensis*[5] and *Shigella*[6]. Studies that further clarify the mechanisms of mortality reduction from azithromycin could help identify complementary interventions or identify alternative interventions that lead to similar benefits but have lower potential to select for antimicrobial resistance[7].

In this pre-specified, secondary analysis of the MORDOR Niger trial, our objectives were to assess the effect of biannual mass distribution of azithromycin to preschool-aged children on serological measures of malaria, bacterial, and protozoan infections. Using a multiplex bead assay, we measured IgG responses to malaria parasites (*Plasmodium falciparum*, *Plasmodium vivax*, *Plasmodium malariae*, *Plasmodium ovale*), *Campylobacter* spp., enterotoxigenic *Escherichia coli* (ETEC), *Vibrio cholerae*, non-typhoidal *Salmonella* (serogroups B and D), *Streptococcus pyogenes* (serogroup A), *Cryptosporidium parvum*, and *Giardia duodenalis*. The IgG panel was selected from a larger library of possible antigens that members of our team had previously developed for integrated serologic surveillance in low resource settings and could serve as plausible endpoints for mass azithromycin treatment[8–11]. The *P. falciparum*, panel included a mix of antigens known to induce both long-lived (MSP-1, AMA1) and short-lived (GLURP-R0, LSA1, CSP, and HRP2) IgG responses[12,13]. We hypothesized that, compared to placebo, children who received azithromycin would have lower levels of infection and thus lower IgG responses to the pathogens measured. Our rationale was based on evidence that azithromycin has antimicrobial activity against *P. falciparum*, *P. vivax*, group A *Streptococcus*, and gram-negative bacteria including *Campylobacter*, ETEC, cholera, and *Salmonella*[14–21]. Although not a first line treatment, azithromycin has antimicrobial activity against enteric protozoans *Cryptosporidium*[22] and *Giardia*[23,24].

## Results

**Study population and setting**. Thirty rural communities were randomized 1:1 to biannual mass azithromycin distribution or placebo offered to all children 1–59 months (NCT02048007). The antibody substudy enrolled 3814 children aged 1–59 months and tested a total of 5642 blood specimens through the 36-month follow-up between March 2015 and June 2018 (Fig. 1). Enrolled children comprised a random sample of up to 40 from each community at each visit and had the same age structure as the overall trial. One community withdrew from the trial at 36 months due to internal politics and trial fatigue. Except for the 6-month measurement, which took place after the seasonal rains, all other specimens were collected from March–July toward the end of the dry season (Supplementary Fig. 1). Treatment coverage was high throughout the study, ranging from 71 to 93% of eligible children (Fig. 1). At baseline, study arms were balanced across demographic characteristics and had similar seroprevalence to measured pathogens (Table 1). As reported in previous studies[9,25], antibody responses to ETEC LTB and cholera CTB

were highly cross-reactive (Supplementary Fig. 2). We excluded cholera CTB results from the primary analyses because we assumed that most of the responses reflected exposure to ETEC based on recent estimates of diarrhea etiology in nearby West African studies[26]. We measured IgG responses using a multiplex bead assay on the Luminex platform (details in Methods). We compared groups using geometric mean IgG responses, seroprevalence, and the seroconversion rate, including measurements at all follow-up times (6, 12, 24, 36 months). These were pre-specified, secondary outcomes for the trial.

**Effects on malaria antibody response**. Children had high IgG seroprevalence to *P. falciparum* MSP-1$_{19}$ and AMA1 antigens, evidence of limited exposure to *P. malariae*, and evidence of very low exposure to *P. vivax* or *P. ovale* (Fig. 2a). A moderate correlation between malarial antibody responses suggests that *P. vivax* or *P. ovale* responses might reflect some limited cross-reactivity from *P. falciparum* infections (Supplementary Fig. 2). However, co-infection with multiple malaria species in this highly endemic region cannot be ruled out. There was heterogeneity between study communities in seroprevalence to longer-lived MSP-1$_{19}$ and AMA1 antibodies, and overall seroprevalence to shorter-lived *P. falciparum* antibodies (GLURP-R0, LSA1, CSP, and HRP2) was considerably lower. IgG responses by age exhibited a characteristic pattern of waning up to 12 months, due to loss of maternal antibodies, with monotonic increases thereafter (Supplementary Fig. 3).

Children who received azithromycin had a transient reduction in *P. falciparum* IgG seroprevalence between ages 12 and 36 months (Fig. 2b), but the overall difference between groups from ages 12 to 59 months was small (−4% difference, 95% CI: −12% to 2%; *P* = 0.32). This corresponded to a 12% relative reduction in the hazard of seroconversion to any of the *P. falciparum* antigens (HR = 0.88, 95% CI: 0.62–1.26). Antigen-specific differences between groups showed small, non-statistically significant reductions among children who received azithromycin based on seroprevalence (Fig. 2c) and force of infection measured by the seroconversion rate (Fig. 2d). Antigen-specific, age-seroprevalence curves showed similar overall patterns between groups, with the largest reductions in *P. falciparum* AMA1 (Supplementary Fig. 4), consistent with comparisons of community-level means (Supplementary Fig. 5). An exploratory analysis that grouped antigens into more durable (MSP-1, AMA1) and less durable (GLRUP-Ro, LSA1, CSP, HRP2) responses showed a slightly larger shift in age-seroprevalence curves for longer-lived responses but the relative reduction in seroconversion rate was similar across both sets of *P. falciparum* antigens (Supplementary Fig. 6).

To assess whether serology might be a more sensitive endpoint in lower transmission settings, we conducted an exploratory subgroup analysis (suggested during peer review, not pre-specified) that stratified communities by baseline malaria parasitemia ≤5% (*n* = 17) versus >5% (*n* = 13). There was modest evidence for a larger reduction in IgG seroprevalence among communities with baseline parasitemia ≤5% (difference = −11%, 95% CI: −22% to 2%) with no difference between groups among communities with higher baseline parasitemia (Supplementary Fig. 7).

**Effects on bacterial and protozoan antibody response**. *Campylobacter* and ETEC seroprevalence were >90% among children 6–24 months, with little heterogeneity in seroprevalence between communities (Fig. 3a). *Salmonella* serogroups B and D, *Streptococcus* serogroup A, *Cryptosporidium*, and *Giardia* seroprevalence was lower compared to the highest transmission pathogens and

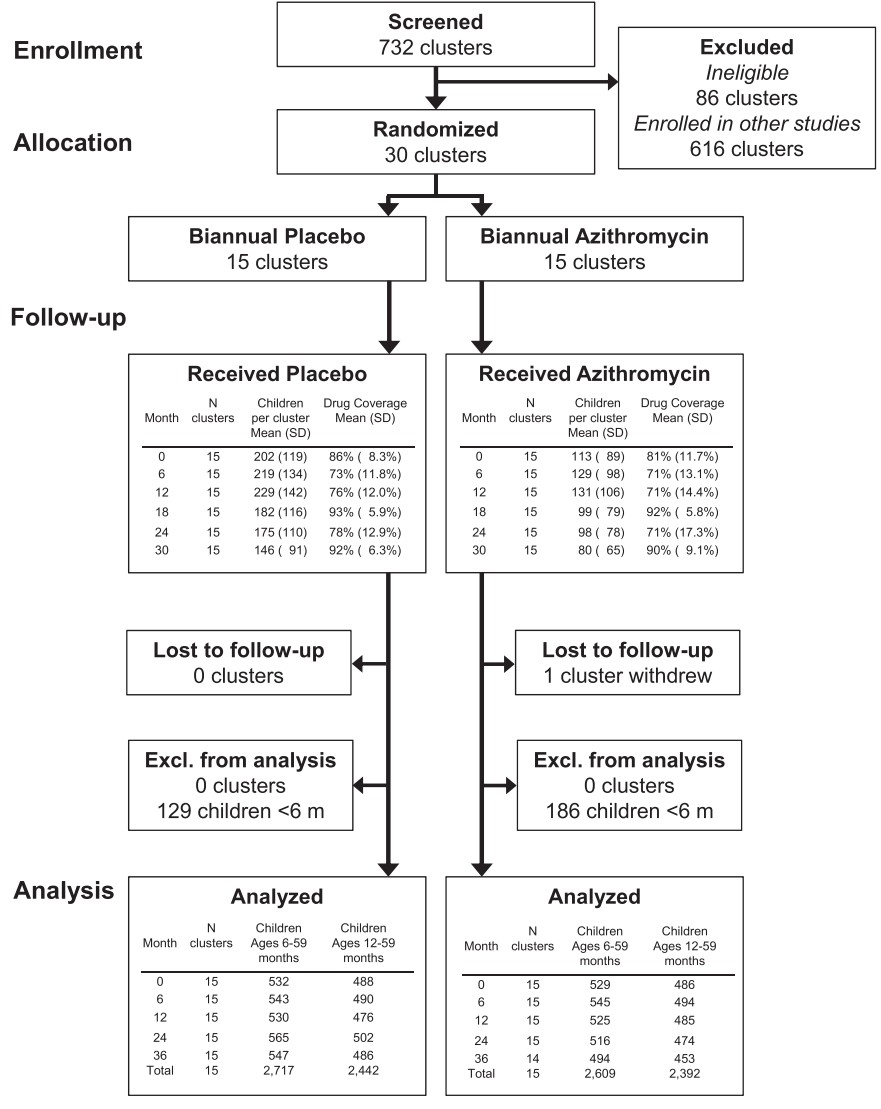

**Fig. 1 Study participant flow.** Thirty community clusters were randomly selected from a larger pool of clusters enrolled in the overall MORDOR Niger trial. A random sample of 40 children ages 1–59 months per community were selected for serological monitoring. Children <12 months old (malaria antigens) and <6 months old (other pathogens) were excluded from the primary analyses based on a pre-specified criteria to exclude maternal IgG contributions. Created with notebook https://osf.io/utxv3.

more heterogeneous between communities (Fig. 3a). Many bacterial and protozoan pathogens showed evidence of maternal IgG contributions through age six months, and mean *Campylobacter* and *Giardia* IgG levels declined modestly beyond ages 18–24 months (Supplementary Fig. 8).

*Campylobacter* seroprevalence was lower among children who received azithromycin compared with placebo (91% vs. 94%, difference = −3%, 95% CI: −5% to −1%; *P* = 0.03) (Fig. 3c), and age-seroprevalence curves showed a consistent reduction across all ages (Fig. 3b). We estimated 0.54 fewer *Campylobacter* seroconversions per child-year at risk, a 29% relative reduction (1.30 vs. 1.84 seroconversions per year), based on a semiparametric proportional hazards model (HR = 0.71, 95% CI: 0.56–0.89; *P* = 0.004), which remained significant after correction for multiple testing (Fig. 3d). There were no statistically significant reductions in seroprevalence or force of infection for other measured bacterial and protozoan pathogens. Age-seroprevalence curves (Supplementary Fig. 9) and community level means (Supplementary Fig. 10) were consistent with the overall results.

**Additional analyses**. Children included in multiple cross-sectional samples between ages 12 and 59 months contributed to longitudinal analyses of malaria seroconversion (919 children, 2197 measurements). Longitudinal samples from children ages 6–59 months (1038 children, 2516 measurements) contributed to analyses of *Salmonella* and *Streptococcus*, and longitudinal samples among children 6–24 months (313 children, 680 measurements) contributed to analyses of the remaining enteric pathogens. Seroconversion rates were generally higher when estimated longitudinally compared with those estimated from age-structured seroprevalence. Longitudinal analyses demonstrated larger reductions in *P. falciparum* seroconversion rates among children who received azithromycin compared to the primary analysis (e.g., AMA1 seroconversion incidence rate ratio: 0.49, 95% CI: 0.32–0.78), but most comparisons were slightly underpowered given the small size of the longitudinal cohort (Supplementary Table 1). Longitudinal comparisons of bacterial and protozoan pathogens were largely consistent with the primary analysis but were also slightly underpowered. For example, the *Campylobacter* seroconversion rate was 19% lower (IRR: 0.81,

**Table 1 Baseline study group characteristics.**

|  | Placebo[a] $N = 15$ | Azithromycin[a] $N = 15$ |
|---|---|---|
| *Age, %* |  |  |
| 0 y | 13 | 12 |
| 1 y | 14 | 14 |
| 2 y | 18 | 20 |
| 3 y | 26 | 24 |
| 4 y | 30 | 30 |
| Female, % | 45 | 48 |
| *Malaria seroprevalence, %[b]* |  |  |
| *P. falciparum* MSP-1$_{19}$ | 79 | 76 |
| *P. falciparum* AMA1 | 67 | 61 |
| *P. falciparum* GLURP-Ro | 26 | 21 |
| *P. falciparum* LSA1 | 11 | 8 |
| *P. falciparum* CSP | 5 | 4 |
| *P. falciparum* HRP2 | 2 | 2 |
| *P. malariae* MSP-1$_{19}$ | 13 | 12 |
| *P. ovale* MSP-1$_{19}$ | 2 | 3 |
| *P. vivax* MSP-1$_{19}$ | 1 | 2 |
| *Bacteria and protozoa seroprevalence, %[c]* |  |  |
| *Campylobacter* sp. p18 or p39 | 92 | 91 |
| ETEC LTB | 88 | 84 |
| *Salmonella* sp. LPS serogroups B or D | 48 | 45 |
| *Cryptosporidium* sp. Cp17 or Cp23 | 85 | 83 |
| *Giardia* sp. VSP-3 or VSP-5 | 77 | 76 |
| *Streptococcus* sp. serogroup A SPEB | 63 | 60 |

[a]Estimates include 559 children in placebo communities and 555 children in azithromycin communities.
[b]Malaria antibody seroprevalence was estimated among children ages 12–59 months (baseline Placebo $N = 488$, Azithromycin $N = 486$), per the age group used in the primary analysis.
[c]Bacteria and protozoan antibody seroprevalence estimated among children ages 6–24 months (ETEC LTB, baseline Placebo $N = 201$, Azithromycin $N = 202$) or among children ages 6–59 months (all others, baseline Placebo $N = 532$, Azithromycin $N = 539$), per the age groups used in the primary analysis.
**Created with notebook https://osf.io/qeg4u.

95% CI: 0.63–1.05) in the azithromycin group compared to placebo in the longitudinal analysis (Supplementary Table 1). Longitudinal analyses showed seroreversion across all antibodies measured except for responses to *Campylobacter* and ETEC, with some evidence for higher *Cryptosporidium* and *Giardia* seroreversion rates in the azithromycin group (Supplementary Table 2).

There was no evidence for effect modification by study phase (Supplementary Fig. 11) or child age at the trial start (Supplementary Fig. 12). Intra-class correlation (ICC) estimates varied across antibodies, with higher between-community standard deviation and higher ICCs for longer-lived malaria responses and *Streptococcus* group A (Supplementary Table 3). At the community level, seroprevalence was correlated with thick smear malaria parasitemia (Spearman's $\rho = 0.45$, Supplementary Fig. 13). Adjusting seropositivity cutoffs by ±20% did not qualitatively change the results (Supplementary Fig. 14). Leave-one-out sensitivity analyses showed overall estimates were unbiased and no single community had undue influence on the analysis (Supplementary Fig. 15).

## Discussion

In this analysis of IgG antibody response to malarial, bacterial, and protozoan pathogens we found that biannual mass distribution of azithromycin to children 1–59 months reduced *Campylobacter* seroprevalence and seroconversion rates. Overall, azithromycin had limited effects on antibody-based measures of pathogen exposure against a backdrop of very high transmission for most pathogens. Antibody responses rose quickly by age and showed that infection was extremely common for most pathogens studied, for example, seroprevalence to *Campylobacter* and ETEC was close to 100% by age 18 months and was >90% for *P. falciparum* by age 40 months. This seroepidemiologic pattern suggests that the intervention failed to reduce overall community transmission[27], and that any effects were most likely due to treatment of existing infections.

A reduction in IgG responses to *Campylobacter jejuni* among children who received azithromycin is consistent with the trial's earlier report of reduced mortality in azithromycin-treated communities attributed to dysentery[3]. The results are also consistent with a separate metagenomic deep sequencing analysis of rectal swabs collected from children in the same communities, which showed a reduction in *Campylobacter upsaliensis* carriage but no other differences in gut microbiome composition among children in azithromycin-treated communities[5]. IgG responses p18 and p39 could reflect previous infections from multiple *Campylobacter* species. A rabbit polyclonal antibody raised against recombinant p18 antigen cross-reacted with an 18-kDa protein in cell extracts from a broad selection of other *Campylobacter* species[28], and the high level of amino acid sequence identity predicted by a Basic Local Alignment Search Tool for Proteins (BLASTP) analysis[29,30] of the *C. jejuni* and *C. upsaliensis* p18 and p39 antigens (88% and 80%, respectively) suggests that measured IgG responses could reflect the previous infection by one or both species (Supplementary Table 4). Although we estimated that azithromycin treatment reduced the hazard of *Campylobacter* seroconversion by 29%, by age 18 months more than 95% of children in both groups were seropositive to *Campylobacter*, reflecting the very high force of infection. These results are consistent with eight, high-resolution, longitudinal cohorts that found 85% of children in low-resource settings had experienced at least one *Campylobacter* infection by age 12 months[31]. *Campylobacter* infections among preschool-aged children have significant clinical sequelae including acute bloody diarrhea (identified as the leading cause)[32], increases in intestinal inflammatory markers[31], and subsequent growth failure[33]. As we describe below, features of this study could have led us to underestimate effects of the intervention on pathogen carriage, so effects demonstrated on *Campylobacter* force of infection despite built-in conservatism in the estimates suggest a potentially important mechanism for the intervention's effect on mortality in Niger. This result motivates the further study of optimal treatment and primary prevention measures to reduce *Campylobacter* transmission, which is a persistent challenge. For example, intensive intervention trials of improved household water, sanitation, and handwashing conditions have failed to reduce infection among children[34–36].

The multiplex assay included a diverse panel of antigens but did not cover all clinically relevant bacterial pathogens that contribute to child mortality, such as *Shigella*[6] or those that cause pneumonia. It is also possible that azithromycin had modest effects on infection with malaria and bacterial pathogens beyond *Campylobacter*, but that IgG responses were an insensitive outcome measure. Azithromycin has broad-spectrum activity against gram-positive and atypical bacteria and has been shown to be highly effective for the treatment of bacterial enteric pathogen infections, including enterotoxigenic *Escherichia coli* (ETEC), *Vibrio cholerae*, and non-typhoidal *Salmonella enterica*[19–21]. Several trials have demonstrated that azithromycin has comparable efficacy to penicillin in treating group A streptococcal pharyngitis[18]. Azithromycin has been shown to have modest antiparasitic activity against *Cryptosporidium parvum*[22]. Less is known about its effect on *Giardia duodenalis* infections in

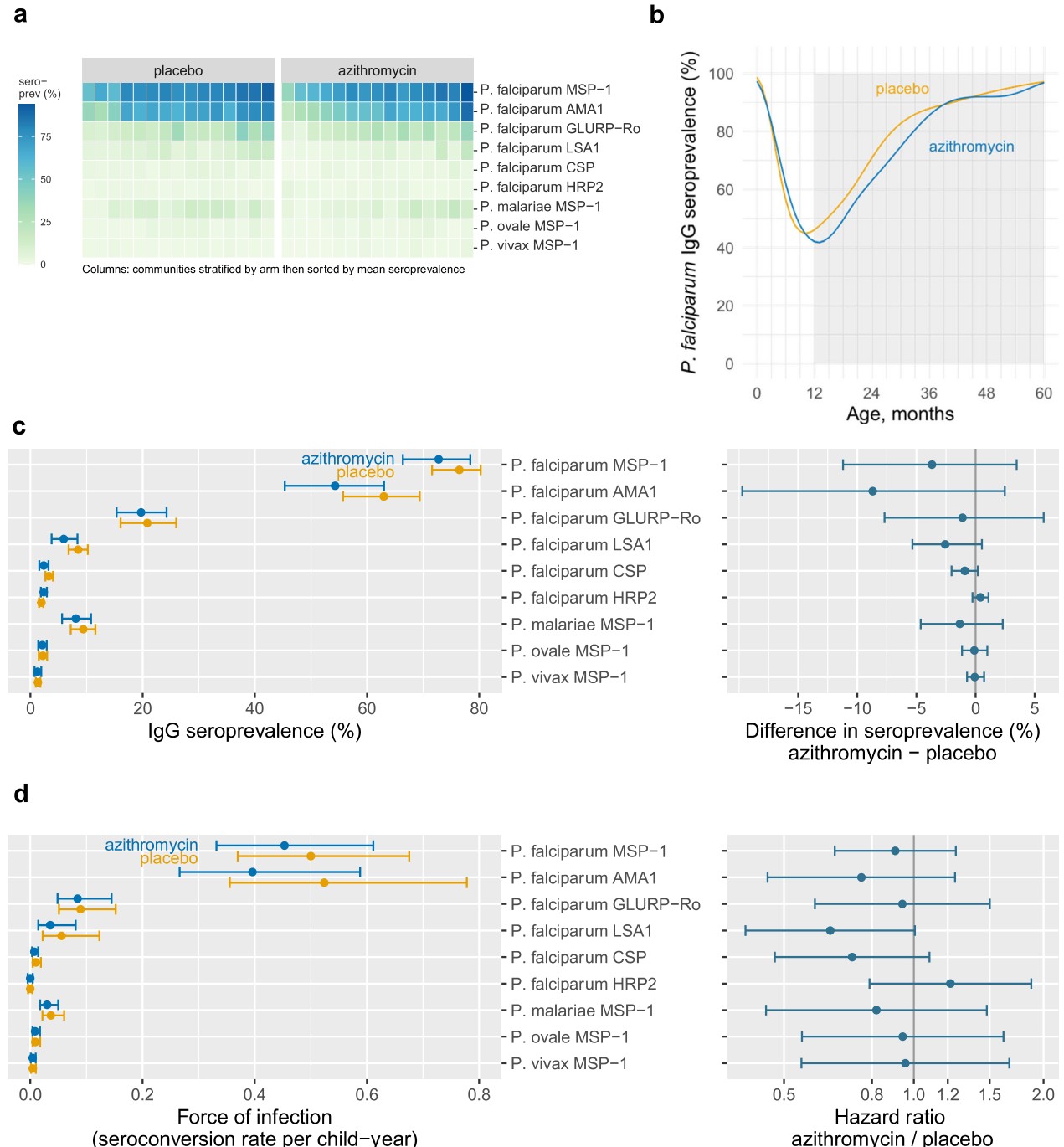

**Fig. 2 Malarial IgG antibody responses by treatment group.** Estimates from 3860 children ages 12–59 months in the MORDOR Niger trial, 2015–2018.
**a** Community-level seroprevalence to malarial antigens. Columns represent individual communities, which were stratified by treatment group and then sorted by overall mean seroprevalence. **b** Mean IgG seroprevalence to *P. falciparum* (positive to any measured antigen) by age and treatment group (lines), estimated with semiparametric splines. The shaded region from 12 to 59 months indicates the age range included in the primary analysis. **c** Antigen-specific IgG seroprevalence by treatment group and difference between groups. Points indicate group means and mean the difference between groups, error bars indicate 95% confidence intervals. **d** Antigen-specific force of infection estimated by the seroconversion rate, and hazard ratio for comparison between groups. Points indicate group means and the hazard ratio between groups, error bars indicate 95% confidence intervals. No between-group comparisons were statistically significant at the 95% confidence level after false discovery rate correction. Created with notebooks https://osf.io/b2v3r, https://osf.io/37ybm, https://osf.io/fwxn5, which include detailed point estimates and additional, consistent results based on geometric mean IgG levels.

humans, but in vitro and in vivo animal studies have shown azithromycin had comparable antimicrobial activity against *G. duodenalis* to metronidazole (the first-line treatment)[23] and small scale (uncontrolled) human studies suggest cure rates in excess of 90%[24]. Azithromycin has modest activity against *P. falciparum* through action against the parasite's apicoplast[14,15], and high efficacy against *P. vivax* both as a prophylactic to prevent infection[16] and as a treatment therapy[17]. Even if azithromycin successfully treated infections from these pathogens, IgG responses might be an insensitive marker of efficacy if the

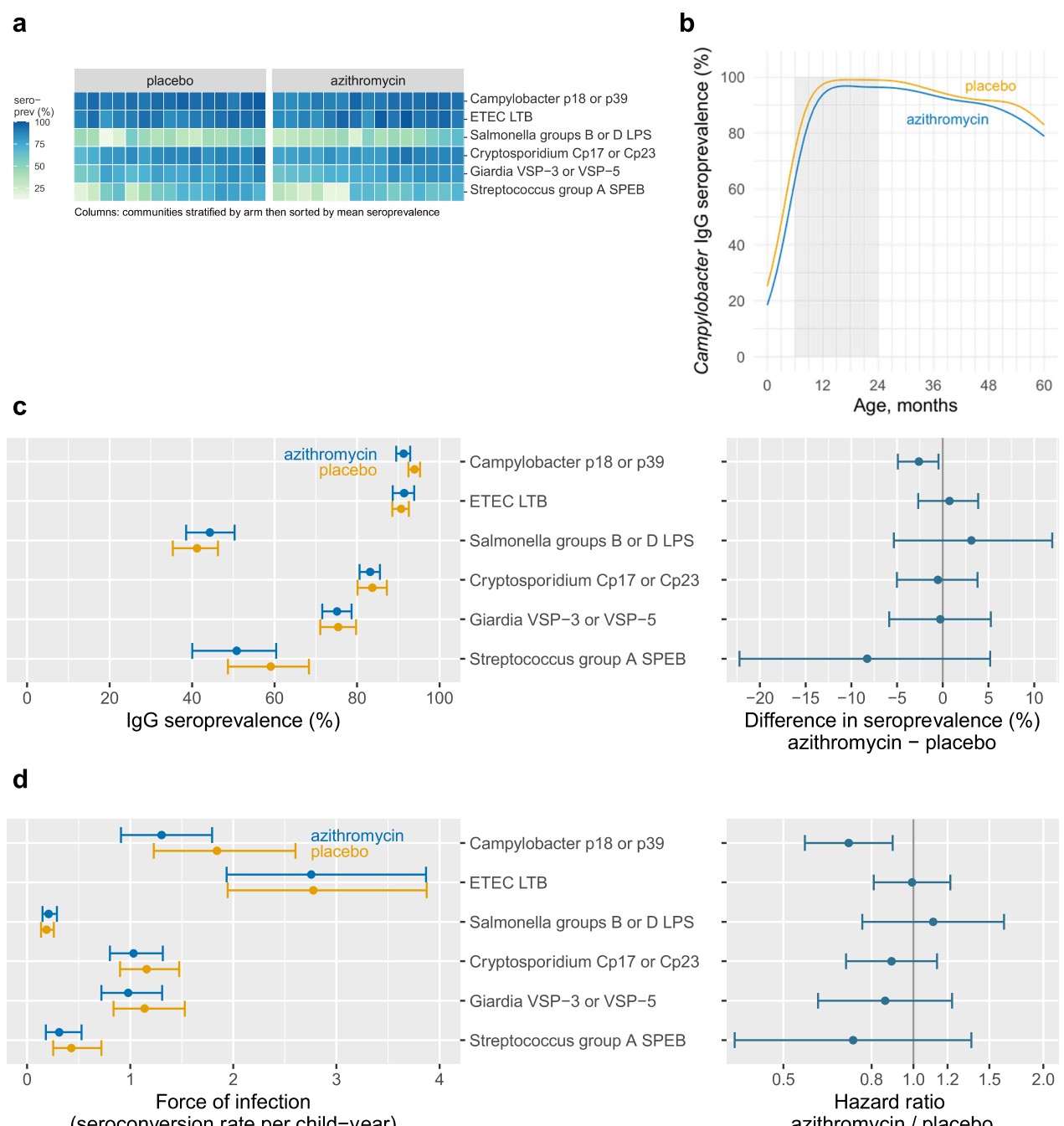

**Fig. 3 Bacterial and protozoan IgG antibody responses by treatment group.** Estimates from 4265 children ages 6–59 months in the MORDOR Niger trial, 2015–2018. **a** Community-level seroprevalence to bacterial and protozoan antigens. Columns represent individual communities, which were stratified by treatment group and then sorted by overall mean seroprevalence. **b** IgG seroprevalence to *Campylobacter* spp. p18 or p39 antigens by age and treatment group (lines), estimated with semiparametric splines. The shaded region from 6 to 24 months indicates the age range included in force of infection analyses, based on a pre-specified rule (*n* = 1496). **c** Pathogen-specific IgG seroprevalence by treatment group and the difference between groups (*n* = 4265 children ages 6–59). Points indicate group means and mean the difference between groups, error bars indicate 95% confidence intervals. **d** Pathogen-specific force of infection estimated by the seroconversion rate, and hazard ratio for comparison between groups. Antibody responses were measured from 1496 children ages 6–24 months (*n* = 4265 children ages 6–59 months for *Salmonella* and *Streptococcus*), based on pre-specified age restrictions. Points indicate group means and the hazard ratio between groups, error bars indicate 95% confidence intervals. The reduction in *Campylobacter* spp. seroconversion rate remained statistically significant after false-discovery rate correction. ETEC enterotoxigenic *E. coli*. Created with notebooks https://osf.io/b2v3r, https://osf.io/smwbn, https://osf.io/fwxn5, which include detailed point estimates and additional, consistent results based on geometric mean IgG levels.

treatment did not reduce the IgG response. For example, IgG longevity or low-density infections that elicited a robust adaptive immune response could explain the small reductions observed in *P. falciparum* IgG responses despite lower levels of parasitemia and parasite density in the azithromycin group[4]. Longitudinal IgG measures among infected children who receive treatment would be needed to definitively assess whether IgG levels drop the following treatment. IgG responses to *Chlamydia trachomatis* dropped only slightly after azithromycin treatment among children in Tanzania[37], but similar studies have not been done for the antigens included in this analysis. Exploratory analyses that stratified by baseline malaria parasitemia showed more consistent reductions in *P. falciparum* seroprevalence in lower transmission communities only (Supplementary Fig. 7). Based on these observations, we surmise that IgG responses might be more useful for measuring the effects of interventions that prevent incident infections or interventions that treat infections in settings without frequent reinfection.

This study had limitations, all of which could have led us to under-estimate effects of azithromycin on pathogen-induced IgG response. Most samples were collected at the end of the dry season, many months after peak malaria season, which coincides with seasonal rains (Supplementary Fig. 1). Subgroup analyses by study phase compared groups at 6-months after enrollment, immediately following the rainy season, and showed no consistently larger effect across antigens at that time point (Supplementary Fig. 11), suggesting that measurement timing alone would not explain the absence of effects on malaria and other pathogens. A second limitation was that the study collected blood spots approximately 6 months after each treatment administration. If azithromycin had short-term effects on infection and antibody response, without effects on longer-term pathogen carriage, then this study could have missed such transient effects. The timing of monitoring visits was chosen to provide longer-term data about community antibiotic resistance (i.e., resistance that persisted even months after the mass drug administration)[7]. However, azithromycin would be expected to have its strongest effect within days or weeks of administration. An analysis of the timing of mortality in MORDOR Niger suggested that most of the intervention effect occurred within three months of distribution[38]. Among children ages 0–24 months, macrolides have been shown to reduce the risk of *Campylobacter* detection in stool significantly 0–15 days after treatment with reduced protection through 45 days, and no protection thereafter[31]. An analysis of enteric pathogen antibody dynamics among children suggested IgG half-life estimates in the range of 10 weeks, implying time to seroreversion of approximately one year absent repeated infections[9]. Thus, the trial could have missed transient effects of azithromycin on pathogen infection, even if clinically significant with respect to child survival, without sustained reductions in community-level transmission because of the slow waning of IgG responses following infection or IgG boosting after new infections between visits. Longitudinal, molecular measures of infection immediately before and within 1–2 weeks of treatment might provide a more definitive measure of azithromycin's effect on pathogen infection. A third limitation was that the primary analysis followed the repeated cross-sectional design and thus did not account for high levels of between-child variability in IgG response. Repeated cross-sectional sampling enabled the trial to continually enroll young children as the study progressed and was protected from potential bias through loss-to follow-up. Nevertheless, longitudinal analyses on a subset of the children who provided multiple specimens over time had similar precision as the primary analysis, illustrating much higher efficiency of longitudinal designs for IgG responses. Despite this limitation, high levels of agreement between the repeated cross-sectional and longitudinal analyses provided a valuable internal consistency check. Finally, the campaign-style distribution of azithromycin every 6 months meant that some children would have been up to 6-months old before their first treatment. For very high transmission pathogens, it is likely that many children had already been infected and may have had a robust IgG response that was insensitive to reduced pathogen carriage following azithromycin treatment.

In conclusion, this study demonstrated that biannual mass distribution of azithromycin reduced antibody-based measures of *Campylobacter* infection, consistent with independent metagenomic analyses in the same study communities. There was no evidence for significant reductions in IgG antibody-based measures of infection with malaria or other measured bacterial and protozoan pathogens. Given the clinically significant sequelae from *Campylobacter* infection among preschool-aged children, our results point to at least one important mechanism through which azithromycin plausibly reduced mortality in Niger. Studies of infection in the weeks immediately following treatment may provide additional insights into the mechanisms through which mass distribution of azithromycin reduces child mortality.

## Methods

**Ethics statement**. The trial protocol was reviewed and approved by the Committee on Human Research at the University of California, San Francisco, and the Niger Ministry of Health's Ethical Committee. Parents or guardians of enrolled children provided oral consent before each azithromycin or placebo treatment and at each specimen collection visit. Parents or guardians were instructed to report any adverse event within 7 days of treatment by contacting their village representative, who then reported events to the site coordinator and UCSF. An independent Data and Safety Monitoring Committee provided additional oversight. Centers for Disease Control and Prevention (CDC) researchers had access to de-identified samples for analysis (no personally identifying information).

**Study design and participants**. MORDOR Niger was a cluster-randomized, placebo-controlled trial that randomized at the community level because of the intervention's campaign-style, biannual mass distribution. Communities with 200–2000 inhabitants based on the Niger 2012 census were eligible for inclusion in the trial, and children ages 1–59 months who weighed >3.8 kg were eligible for treatment. An intensive morbidity monitoring trial enrolled 30 communities and randomized them 1:1 to receive either biannual azithromycin or placebo to all children 1–59 months old (NCT02048007). The trial used a repeated cross-sectional design, whereby 40 children per community were randomly sampled in each measurement round and invited to participate in a monitoring visit. The trial's open cohort design meant that children aged in and out of the study based on their age at the time of treatment. Field staff collected dried blood spots from participating children at baseline and annually thereafter at 12, 24, and 36 months of follow-up. The antibody substudy included a supplemental visit at 6 months, following the malaria season. Children who were randomly selected in multiple survey rounds contributed to longitudinal analyses. Field data were collected using handheld tablets (Android operating system version 5) and a custom application designed for the study (versions 2–4, Conexus Inc., Los Gatos, CA), which encrypted and transmitted the data to a secure server hosted by Salesforce.com.

**Randomization and masking**. Communities were randomized 1:1 using a sequence the trial biostatistician (TCP) generated. Unmasked members of the data team and Pfizer labeled the study drugs. Placebo and azithromycin had identical packaging to maintain masking. Participants, field staff, laboratory staff, analysts, and all investigators were masked to treatment assignments throughout the trial. Masked analyses were completed using a shuffled version of the treatment assignment variable[39]. Data were unmasked only after the final table and figure shells had been populated (documented through the article's GitHub repository).

**Procedures**. Children ages 1–59 months received azithromycin or identically-appearing placebo at the time of enrollment and every 6 months over the course of the study through community-wide census and MDA distributions performed by study staff. Children were given a volume of oral suspension equal to at least 20 mg per kilogram of body weight, which was measured by hanging scale for children unable to stand or by height stick for children who could stand, consistent with the Niger trachoma program. Children with a known allergy to macrolides did not receive azithromycin or a placebo.

## Antibody testing

*Sample collection and preparation.* Dried fingerprick blood spots (DBS) were collected onto calibrated filter paper wheels with six 10 µl extensions (TropBio Pty Ltd., Townsville, Queensland, Australia), which were dried and packaged in individual sealable plastic bags with desiccant and stored in a −20 °C freezer prior to shipping to CDC. DBS were shipped to CDC at ambient temperature and stored in a −20 °C freezer[40]. A single 10 µl extension per participant was eluted overnight at 4 °C in phosphate-buffered saline (PBS) containing 0.5% casein, 0.3% Tween-20, 0.5% polyvinyl alcohol, 0.8% polyvinylpyrrolidone, 0.02% sodium azide, and 3 µg/mL *E. coli* extract (Buffer B). Elutes were diluted to a final concentration of 1:400 with additional Buffer B to test on the multiplex bead assay (below).

*Antigen coupling.* Antigens were covalently coupled to polystyrene beads (SeroMap Beads; Luminex Corporation, Austin, TX) by modifying carboxyl groups on the beads to ester groups using 1-ethyl-3-(3-dimethylaminopropyl) carbodiimide (EDC) (EMD Millipore Calbiochem, USA) in the presence of N-hydroxysulfosuccinimide (sulfo-NHS) (Thermo Scientific Pierce, USA)[40]. Ester groups on the beads bind to primary amine groups on antigens to create stable amide covalent bonds. Beads were briefly sonicated in a water bath and washed with 0.1 M sodium phosphate buffer, pH 6.2 (NaP) in preparation for bead activation. Beads were protected from light and rotated for 20 min in NaP with 5 mg/ml each EDC and NHS. After activation, activated beads were washed and suspended in coupling buffer, antigen added, and rotated at room temperature for 2 h. Coupling buffers and antigen amounts were previously determined for each antigen (Supplementary Table 5). After 2 h, antigen-coupled beads were washed with PBS and unreacted sites were blocked with 1% bovine serum albumin (BSA) in PBS for 30 min. Antigen-coupled beads were resuspended in storage buffer (PBS, 1% BSA, 0.05% Tween-20, 0.02% NaN₃, protease inhibitors) and kept at 4 °C until use in assays.

*Antigen panel.* The panel included malaria antigens to various stages of *P. falciparum* infection, including sporozoite (CSP), hepatocyte (LSA1), merozoite (GLURP-R0) and erythrocyte (MSP-1₁₉, AMA1, HRP2), with MSP-1₁₉ and AMA1 thought to induce longer-lived IgG responses compared with other included *P. falciparum* antigens[12]. Species specific MSP-1₁₉ was used to detect *P. vivax, P. malariae, P. ovale*[10]. Bacterial and protozoan antigens from *Campylobacter jejuni* (p18, p39), enterotoxigenic *Escherichia coli* labile toxin B subunit (ETEC LTB), *Vibrio cholerae* toxin B subunit (CTB), *Salmonella* serogroups B and D (LPS), *Cryptosporidium parvum* (Cp17, Cp23), *Giardia duodenalis* (VSP-3, VSP-5), and *Streptococcus pyogenes* serogroup A Pyrogenic Exotoxin B (SPEB) were also coupled to beads[9,41].

*Multiplex bead assay.* We measured IgG responses using a multiplex bead assay on the Luminex platform. Antigen-coupled beads (1250 per well/bead coupling) were incubated in 96-well assay plates with diluted sample for 1.5 h then washed with 0.3% Tween-20 in PBS (PBST). Beads incubated with biotinylated mouse anti-human IgG (1:500 dilution) and biotinylated mouse anti-human IgG4 (1:1250 dilution) for 45 min to detect IgG bound to the beads. After additional washes with PBST, beads were incubated for 30 min with phycoerythrin-labeled streptavidin (1:200 dilution) to detect bound biotinylated anti-human IgG. After detection, beads were washed with PBST and incubated for 30 min with PBS containing 0.5% BSA, 0.05% Tween-20, and 0.02% sodium azide to remove loosely bound antibodies. After a final wash with PBST, beads were resuspended in PBS and stored at 4 °C overnight. The next day, assay plates were read on a Bio-Plex 200 instrument (Bio-Rad, Hercules, CA) equipped with Bio-Plex manager 6.0 software (Bio-Rad). The median fluorescence intensity (MFI) with the background from the blank well (Buffer B alone) subtracted out (MFI-bg) was recorded for each antigen for each sample. DBS samples were masked and randomly ordered by the UCSF trial coordinating center before sending them to the laboratory at the CDC. Samples were run on 63 plates and each plate included positive controls from a high positive sera pool (1:400 dilution), a low positive sera pool (1:6400 dilution), and a normal human sera (single) control. To assess plate-to-plate variation, we estimated the standard deviation (SD) for each antigen's responses across plates. The laboratory protocol specified that a plate should be re-analyzed if more than half of the antigens had SDs that deviated by >20% from the overall average in multiple controls. Two of 63 plates failed this criterion for one control but passed in the other two controls and were thus not repeated. For the positive control sample responses to the 22 antigens used in this study, the average CV% was 10.0 with an SD of 4.9. The median CV% was 9.3 with a range of 2.9–20.2%.

## Malaria parasitemia

At the time of dried blood spot collection, a drop of blood was used to prepare a thick smear slide that was stained with 3% Giemsa and later assessed for the presence and density of malaria parasites[4]. Lab personnel were masked to specimen groups, and two independent team members at the Centre de Recherche Médicale et Sanitaire (Niamey, Niger) with experience in parasitemia assessment recorded the presence and density of parasites. Any discrepancies were adjudicated by a third, masked, independent laboratory worker until they reached a majority consensus. We estimated community-level malaria parasitemia as the proportion of children with positive thick smears by microscopy.

## Outcomes

We compared groups using geometric mean IgG responses, seroprevalence, and the seroconversion rate, including measurements at all follow-up times (6, 12, 24, and 36 months). These were pre-specified, secondary outcomes for the trial (NCT02048007).

## Statistics

*Seropositivity cutoffs.* We log₁₀ transformed Luminex MFI-bg IgG levels before analysis. We converted IgG responses to seropositive and seronegative classes using seropositivity cutoffs derived from the mean plus 3 SD of responses from a panel of 92 anonymous, adult, USA resident blood donors (all malaria antigens), from ROC-derived cutoffs based on responses from known positive and negative specimens from North America (*Cryptosporidium* n = 68, *Giardia* n = 32)[42], or from the mean plus 3 SD of presumed unexposed measurements (all other antigens). We identified presumed unexposed measurements as those collected among children ≤12 months old that preceded a 10-fold increase in IgG in the longitudinal subsample[9]. For pathogens with multiple antigens measured, we classified children as seropositive if they were positive to any antigen. For *P. falciparum*, we examined individual antibody endpoints as well as a composite outcome, defined as a seropositive response to any *P. falciparum* antigen measured. An exploratory analysis (not pre-specified) grouped *P. falciparum* responses by antigens with more durable IgG responses (MSP-1, AMA1) and less durable IgG responses (GLRUP-Ro, LSA1, CSP, HRP2).

*Age restrictions.* We restricted the age ranges included in the analyses based on pre-specified rules to exclude maternal IgG contributions and to focus the analysis on age ranges with heterogeneity in IgG responses. Before data were unmasked, we examined age-antibody profiles for each antigen and excluded from the primary analyses measurements among children <12 months (malaria responses) and <6 months (bacterial and protozoan responses) to remove potential maternally derived IgG contributions (SI Fig. 3, SI Fig. 8)[43]. Additionally, we limited all analyses of ETEC LTB to ages 6–24 months and force of infection analyses based on seroconversion rates to measurements among children ≤24 months (all enterics except *Salmonella* sp.) because nearly all children older than 24 months were seropositive.

*Descriptive summaries at the community level.* We summarized the SD of community-level seroprevalence and estimated the ICC for community-level responses using a mixed-effects binomial model with a parametric bootstrap to estimate 95% confidence intervals for the ICC (1000 iterations)[44]. We compared community-level malaria parasitemia prevalence and seroprevalence to *P. falciparum* antigens using Spearman rank correlation.

*Estimation of mean differences.* All comparisons were intention-to-treat. We compared mean differences between groups in geometric mean IgG levels, seroprevalence, and malaria parasitemia by pooling all post-treatment measurements. We estimated 95% confidence intervals using a non-parametric bootstrap that resampled communities with replacement (1000 iterations). We calculated exact permutation P-values from the randomization distribution of mean differences.

*Age structured seroprevalence and force of infection.* We used a current status, semi-parametric proportional hazards model to estimate the force of infection from age-structured seroprevalence[45]. We fit a generalized additive mixed model with binomial errors and complementary log–log link

$$\log\left(-\log\left[1 - P\left(Y_{ij} = 1 | A_{ij}, X_i, b_i\right)\right]\right) = g\left(A_{ij}\right) + \beta_1 X + b_i \quad (1)$$

where $Y_{ij}$ is antibody seropositivity, $A_{ij}$ is the age for child $j$ in community $i$. $X_i$ is treatment allocation for community $i$ (equal to 1 for azithromycin, 0 for placebo). The model included community-level random effects, $b_i$, to allow for correlated outcomes within the community. Function $g(\cdot)$ was parameterized with cubic splines that had smoothing parameters chosen through generalized cross-validation using the default in the R mgcv package[45]. The primary analysis pooled information over all post-randomization measurements available at the time of analysis (months 6, 12, 24 and 36). We estimated the hazard ratio (HR) of seroconversion associated with the biannual mass distribution of azithromycin as $\hat{\theta}_{HR} = \exp(\hat{\beta}_1)$.

We estimated age- and treatment-specific seroprevalence from the model as

$$\hat{P}(Y = 1 | A = a, X = x) = 1 - \exp\left(-\exp[\hat{\eta}(a, x)]\right) \quad (2)$$

and we estimated age- and treatment-specific force of infection from the model as

$$\hat{\lambda}(a) = \eta'(a, x)\ \exp[\hat{\eta}(a, x)] \quad (3)$$

where $\eta'(a, x)$ is the first derivative of the linear predictor from the complementary log–log model, $\eta(a, x)$[46]. We estimated $\eta'-(a, x)$ using finite differences from the model predictions[45,47]. We estimated approximate, simultaneous 95% confidence intervals around age-specific seroprevalence and age-specific force of infection curves with a parametric bootstrap (10,000 replicates) from posterior estimates of the model parameter covariance matrix[48]. We used the age-specific force of infection curves to visually confirm proportional hazards between groups. To

estimate the marginal average force of infection in each group, we integrated overage[49].

In some cases, the same child was measured multiple times during the trial and contributed multiple measures within a community over the course of follow-up (next section). We considered the inclusion of a child-level random effect in the age-structured seroprevalence model, nested within the community, to model this additional potential source of outcome correlation. However, models that included this more complex random effects structure failed to converge in most cases. Since our target of inference was the HR for azithromycin-treated communities versus placebo, inclusion of a community-level random effect, the independent unit of analysis and inference, should lead to unbiased estimates of the standard error even without attempting to model additional sources of variation within the community[50].

*Longitudinal rate estimates.* A subset of children was sampled in multiple, repeated cross-sectional surveys and thus provided longitudinal antibody measurements (two to five visits). Children included in multiple cross-sectional samples between ages 12 and 59 months contributed to longitudinal analyses of malaria (919 children, 2197 measurements, median [range] measurements per child 2 [2, 5]). Longitudinal samples from children ages 6 to 59 months (1038 children, 2516 measurements, median [range] measurements per child 2 [2, 5]) contributed to analyses of *Salmonella* and *Streptococcus*, and longitudinal samples among children 6–24 months (313 children, 680 measurements, median [range] measurements per child 2 [2, 4]) contributed to analyses of the remaining enteric pathogens. We conducted a supplemental analysis in this opportunistic subgroup to estimate prospective seroconversion and seroreversion rates. We defined seroconversion as an increase in IgG MFI-bg to a level above the antibody's seropositivity cutoff. For pathogens with multiple measured antigens, a child was deemed to have seroconverted if either antibody response met the definition for seroconversion. We assumed that seroconversions occurred at the midpoint of the interval between measurements when estimating person-time at risk. We jointly estimated seroreversion rates using the same approach, using a decrease in IgG across the seropositivity cutoff. We used a non-parametric bootstrap, resampling clusters with replacement, to estimate 95% confidence intervals for rate and incidence rate ratio estimates.

*Subgroup analyses.* We pre-specified examining treatment differences by age at the trial's start date (<6 months vs. older) and by trial phase (6, 12, 24, and 36). We hypothesized that azithromycin could reduce antibody-based measures of transmission more among younger children who were immunologically naïve, and at later phases of the trial due to additional rounds of biannual MDA. For the age-stratified subgroup, we focused on children 6 months or younger (including not yet born) at the start of the trial because age 6 months is approximately when we would expect children to begin to produce their own IgG response (additional details in the statistical analysis plan)[43]. We omitted a pre-specified subgroup analysis for the rainy versus dry season because we ultimately determined that too few samples were collected after the rainy season (6-month samples only).

An exploratory subgroup analysis stratified of *P. falciparum* IgG comparisons by baseline malaria parasitemia (not pre-specified). We estimated the proportion of children who were malaria positive by thick smear microscopy in the baseline survey and identified a natural break in the distribution with 17 communities ≤5% parasitemia prevalence and 13 communities >5% parasitemia prevalence.

*Sensitivity analyses.* We assessed the sensitivity of the results to the seropositivity cutoff for each antigen by changing cutoff values ±20% and then repeating the primary analyses. We assessed the sensitivity of the results to individual communities by conducting a leave-one-out analysis that examined the distribution of analysis results excluding each community in turn and computed a jackknife bias estimate for the difference in seroprevalence and the log HR of seroconversion rates[51].

*Statistical power, detectable effects, and adjustment for multiple comparisons.* The MORDOR morbidity monitoring trial was designed around the primary antimicrobial resistance monitoring endpoints[5,52]. For the present analyses, assuming a sample of 15 communities per arm and 140 measurements per community over four rounds, the seroprevalence of 65% (*P. falciparum* MSP-1), a community-level ICC of 0.004, and a two-sided alpha of 5%, we estimated that we would have 80% power to detect a reduction of 5.4 percentage points in seroprevalence due to intervention[53] (the pre-analysis plan provides additional details, https://osf.io/d9s4t/), seroprevalence, and ICC assumptions were estimated from the nearby PRET trial)[54]. Within each set of analyses, we estimated *P*-values adjusted for multiple comparisons allowing for a 5% false-discovery rate using the Benjamini–Hochberg correction[55].

**Reporting summary.** Further information on research design is available in the Nature Research Reporting Summary linked to this article.

## Data availability

De-identified individual participant data generated in this study have been deposited in the Open Science Framework (https://osf.io/954bt) and in the Dryad database (https://doi.org/10.7272/Q6VX0DSD)[56]. Results from BLASTP searches, including accession numbers, are summarized in Supplementary Table 4.

## Code availability

All code used in the study is available through GitHub (https://github.com/proctor-ucsf/mordor-antibody) and the Open Science Framework (https://osf.io/954bt). Analyses used R statistical software, version 4.1.1.

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

## Acknowledgements

This work was supported by the Bill & Melinda Gates Foundation (award No. OPP1032340 to T.M.L.) and was supported in part by an unrestricted grant from Research to Prevent Blindness (to Proctor Foundation faculty) and by the National Institute of Allergy and Infectious Diseases (award No. K01-AI119180 to B.F.A.). The Gates Foundation approved the study design but had no role in data collection, data analysis, data interpretation, or writing of the report.

## Author contributions

Following CRediT taxonomy, conceptualization (D.L.M., E.R., T.M.L., J.D.K., and B.F.A.), data curation (J.D.K., V.L., and B.F.A.), formal analysis (B.F.A.), funding acquisition (T.M.L.), investigation (D.L.M., E.R., J.W.P., and B.F.A.), methodology (B.F.A.), project administration (A.M.A., R.M., E.L., and T.M.L.), resources (J.W.P., D.L.M., E.R., and E.B.G.), software (B.F.A., T.C.P., and V.L.), supervision (A.M.A., R.M., E.L., and K.S.O.), validation (E.B.G. and E.R.), visualization (B.F.A.), writing—original draft preparation (A.M.A., B.F.A., D.L.M., and E.B.G.), writing—review and editing (A.M.A., R.M., E.B.G., E.R., J.W.P., E.L., K.S.O., V.L., C.E.O., J.D.K., T.C.P., T.D., T.M.L., D.L.M., and B.F.A.).

## Competing interests

The authors declare no competing interests. The findings and conclusions in this article are those of the authors and do not necessarily represent the official position of the Centers for Disease Control and Prevention. Use of trade names is for identification only and does not imply endorsement by the Public Health Service or by the U.S. Department of Health and Human Services.

## Additional information

**MORDOR-Niger Study Group**

Ahmed M. Arzika[1], Ramatou Maliki[1], Elodie Lebas[4], Kieran S. O'Brien[4,5], Catherine E. Oldenburg[4,5,6], Thuy Doan[4,5], Travis C. Porco[4,5,6], Jeremy D. Keenan[4,5], Thomas M. Lietman[4,5,6] & Benjamin F. Arnold[4,5,7]

A full list of members and their affiliations appears in the Supplementary Information.

