## [Peer Review File · Nature Communications]

Effect of biannual azithromycin distribution on antibody responses to malaria, bacterial, and protozoan pathogens in NigerReviewers' Comments:

Reviewer #2:

Remarks to the Author:

In this study the authors performed a follow-up analysis for the MORDOR trial in Niger, which found that mass distribution of azithromycin to preschool aged children reduced all-cause mortality by 18%. Their objective was to examine potential mechanisms for mortality reduction. They did this by estimating differences in IgG antibody responses to eleven pathogens between azithromycin and placebo groups 6 months after each round of treatment. They found high levels of background transmission and, despite high levels of heterogeneity in IgG responses, they found a small but significant reduction in *Campylobacter* spp. force of infection in the communities that received azithromycin.

This manuscript is well written, the analyses are thoughtfully laid out and implemented, and the presented results support the authors' conclusion that reduction in *Campylobacter* infection may be one of the reasons for reduction in mortality following azithromycin treatment. The two main things that were missing for me in the paper were: (1) potential sources of heterogeneity in the IgG assays and responses, and their implications for inferring a mechanism, and (2) what can be done with this new information going forward. I have described these and other comments in detail below.

Main comments:

1) It's a bit unclear to me whether the signal for *Campylobacter*, as oppose to other pathogens, is due to something biological/epidemiological and/or whether it's a reflection of the assay sensitivity or timing to detect this particular pathogen. The independent metagenomic study supports their conclusions, but I'm wondering if this is something the authors can also interrogate in their serological data. Is there information on how sensitivity to detecting these infections varies or on duration of the bacterial and protozoal antibody responses? Were there differences in between- or within-plate variation for the different antigens in the multiplex bead assay?

Similarly, what are other potential sources of IgG variation between participants? The authors touch on several limitations of their serological approach and the timing of the data collection, but I'm wondering if some additional sources of variation could be examined (and if not here, then in a future study). In particular, differences in malnutrition, whether or not children actually received treatment, access to other health services, and other co-infections or co-morbidities, all of which could impact antibody responses as well as child mortality.

2) The *Campylobacter* finding confirms previous studies and suggests a plausible mechanism for reduction in mortality, but I'm not sure how this information can be used to inform treatment or other interventions. If the authors have additional ideas, it would be helpful to include them. If further study is needed, then it would be helpful to know more specifically what those next steps should be.

Another implication of their results, which the authors include in the discussion, is that for multiplex antibody assays to be useful as trial endpoints, they should be done using detailed longitudinal measurements. This is concrete information that can be used in future studies.

Additional comments:

1) Why was this particular multiplex IgG panel selected? Was it because azithromycin has antimicrobial activity against each of these pathogens? If so, it would be helpful to connect those dots in the introduction.

2) Buffer-only background absorbance levels were subtracted from MFI measurements in the multiplex bead assay. Were any other positive or negative controls or dilution series included on each plate that could be used to measure (and potentially adjust for) plate-to-plate variation?

3) What were the panels of known seronegative samples for malaria, *Cryptosporidium*, and *Giardia*? Were these from this population in Niger or elsewhere? It would be helpful to have more details on this part of the methods. I'm also curious, would moving the seropositivity cutoffs up or down ~20% impact the overall conclusions?

Reviewer #4:

Remarks to the Author:

The work presented in this manuscript is of high significance for infectious disease intervention trials and the application of serological endpoints to aid in study evaluation. The following comments indicate areas where more clarity or details could be provided by the authors.

Results

Study population and setting

1. pg 4 "antibody substudy enrolled 3,814 children...5,642 blood specimens."

o While it is indicated in the caption of Figure 1 that there is a random sample of 40 children from each of the 30 communities in the trial, it would be good to note this here also.

o Additionally, would be good to confirm here that the random sample for the sub-study was representative of the same age distribution as the main study.

2. pg 4 "specimens were collect March – July toward the end of the dry season"

o It would be important to note potential differences in antibody decay rates both between pathogens and between antigens for the same pathogens. It is mentioned briefly in the Discussion, but this could have a large bearing of future selection of markers that are best used for evaluation of intervention effects over short period of time.

3. pg 4 "antibody responses to ETEC LTB and cholera CTB were highly cross-reactive"

o Pf and Pv MSP antigens have also been known to have some degree of cross-reactivity?

Confirmation that this was not the case for those used in this study would be helpful

Effect on malaria antibody response

4. pg 5 "evidence of very low exposure to *P.vivax* or *P.ovale*"

o Related to the issues of Pf/Pv cross-reactivity, the low prevalence here may help indicate lack of substantial cross-reactivity, but on the other hand, sero-prevalence for Pv may be artificially low depending on the negative population used to select seropositivity thresholds and if the negative population used differs between the Pf and Pv antigens. See comments on Methods section on cut-off method clarifications.

5. pg 5 "heterogeneity between study communities in the longer-lived MSP1.19 and AMA1 antibodies, and overall seroprevalence go shorter-lived Pf antibodies"

o This sentence could be reworded so it is clear whether the heterogeneity is referring to variation in village/community level sero-prevalence to the same antigen, or variation in sero-prevalence to different antigens. Or both?

o It would be helpful to cite literature on which MSP/AMA are considered longer-lived and GLURP, LSA, HRP are considered shorter-lived

o A mention that HRP2 is not a Pf antigen, but a by-product of parasite biomass would be good here. It would be useful to mention, if this is measuring Ab responses to circulating HRP in bloodstream, and whether these Ab tend to be longer or shorter-lived than Ab responses to parasite-specific antigens.

6. pg 5 "Antigen-specific, age-seroprevalence curves showed similar overall patterns between groups,

with largest reductions in *P. falciparum* AMA1"

o In supplementary figure SI Figure 1, 4 out of the 6 Pf antigens show an average MFI across all ages that is below the seropositivity threshold, other than MSP1.19 and AMA1. Hence, is the overall seroprevalence result here almost entirely due to MSP and AMA? What added information do the other antigens provide, if any?

Effect on bacterial and protozoan antibody response

7. pg 5 "Campylobacter and ETEC seroprevalence were >90% among children 6-24 months, with little heterogeneity between communities

o Same comment as 5 for malaria above on clarifying what heterogeneity is referred to.

Additional analyses

8. pg 6 "Children included in multiple cross-sectional samples between ages 12-59 months contributed to longitudinal analyses of malaria seroconversion (919 children, 2,197 measurements). Longitudinal samples from children ages 6-59 months (1,038 children, 2,516 measurements) contributed to analyses of Salmonella and Streptococcus, and longitudinal samples among children 6-24 months (313 children, 680 measurements) contributed to analyses of the remaining enteric pathogens."

o What was the median (and range) of measurements per child in each set of longitudinal analyses?

9. pg 6 "most comparisons were slightly underpowered given the small size of the longitudinal cohort"
o Was there a method used here to estimate study power / sample size for this analysis? If not, would be useful to include in the Discussion the challenges in determining correct sampling frame to achieve study power in these type of analyses.

Discussion

10. pg 7 "azithromycin had limited effects on antibody-based measures of pathogen exposure against a backdrop of very high transmission for most pathogens."

o Is this implying that antibody responses and sero-prevalence have saturated for these pathogens in these transmission intensities? Would this be affected by methods used to determine sero-positivity cut-offs, e.g. selection of negative population?

11. pg 8 "Previous results from this trial found that azithromycin reduced mortality attributed to malaria, diarrhea, dysentery, and pneumonia [3], and reduced malaria parasitemia"

o It may be useful to include these non-serological results in supplement as a direct comparison against the IgG data, or to show the degree of correlation between the serological vs. molecular of parasite measures

12. pg 8-9 "our finding of no clear reduction in IgG responses to malaria and bacterial pathogens beyond Campylobacter suggests that in high transmission settings, infrequent IgG measurements by themselves are potentially an insensitive trial endpoint."

o It may likely be the latter – if antigens used have antibody decay rates that are much longer than the time period of evaluation, or do not decay rapidly enough to detect change in an intervention trial. In the case of long-lived Ab responses (or rather Ab responses that maintain a stable MFI for a long period as opposed to Ab responses that wane rapidly but still remain as a low level), more frequent measures may be unlikely to be more useful as a trial endpoint. What is needed are antibody responses that boost and decay rapidly during a period of time that matches trial follow-up.

13. pg 10 "If azithromycin had short-term effects on infection and antibody response, without effects on longer term pathogen carriage, then this study could have missed such transient effects."

o It seems that rather than capturing transient effects, it would be more important to define the duration of protection offered by the interventions. Do the authors think there is potential for longitudinal serological assessment (or at well-defined time points) be able to determine this better than non-serological measures? If so, what might be feasible study designs?

Methods

Study design and participants

14. pg 12 "Children who were randomly selected in multiple survey rounds contributed to longitudinal analyses."

o This would be a good place to include details on the number of children who were sampled in

multiple rounds, and the median/range of measurements per child.

Seropositivity cutoffs

15. pg 15 "from a panel of known seronegative sera (malaria antigens)."

o How was sero-negativity determined? Were these non-endemic malaria naïve samples or samples from endemic areas? Authors should mention in the Discussion if this would cause any differences between sero-prevalence measures across different malaria markers. e.g., were there differences in negative populations used for Pf vs. Pv that led to very low sero-prevalence for Pv?

Age structured sero-prevalence and force of infection

16. pg 16 "The model included community-level random effects to allow for correlated outcomes within community."

o Given that there are children with multiple samples in repeated cross-sectional surveys, was there a restriction on number of samples per individual in the sero-prevalence, FOI and hazard ratio analyses e.g., first sample per individual? Or if all samples per individual, accounting for non-independence/correlation by individual?

Subgroup analyses

17. pg 18 "We hypothesized that azithromycin could reduce antibody based measures of transmission more among younger children who were immunologically naïve and at later phases of the trial due to additional rounds of biannual MDA."

o How was the hypothesized age threshold determined?

Statistical power, detectable effects, and adjustments for multiple comparisons

18. pg 18 "assuming a sample of 15 communities per arm and 140 measurements per community over four rounds, seroprevalence of 65% (*P. falciparum*), a community-level ICC of 0.004, and a two-sided alpha of 5%, we estimated that we would have 80% power to detect a reduction of 5.4 percentage points in seroprevalence due to intervention."

o Was the study powered specifically for Pf, and if so, what is the reason for this as opposed to other pathogens?

o Clarify that this is sero-prevalence to all Pf antigens combined?

o Was the ICC based on village/community-level measure of seroprevalence? If so, it would be useful to report the mean and SD here.

Figures and Tables

19. Table 1. Baseline study group characteristics

o Alongside overall sero prevalence by antigen, here would be a good place to also include community-level seroprevalence (mean, SD across all or by intervention)

o N used for individuals in caption could be changed so it is not confused with N=communities in the columns of the table

Supplementary Material

20. SI Figure 3 – A. specimens collected by calendar month and study phase

o It is difficult to interpret this plot. Are there multiple phases in the same calendar month (column) due to variable enrolment date for individuals/villages?

21. SI Figure 4 – Correlation between antibody responses for ETEC LTB and cholera CTB

o A similar plot for MSP between malaria species would also be useful to confirm there is limited cross-reactivity.

Reviewer #5:

Remarks to the Author:

This study expands on the primary outcome from the MORDOR trial where a significant reduction in all-cause mortality was associated with biannual azithromycin treatment in under 5's, and attempts to identify the basis for this reduction through assessing changes in serological profiles between the two arms. The manuscript is well written, the experiments well performed and analysed. The conclusions

are robust albeit limited in their ability to explain the overall impact of the trial. This may be due to the high transmission setting that could mask some of the studies impact, or be due to the study design (e.g. limited sampling).

Major Comments

- Was any sensitivity analysis performed to assess the impact of removal of one or more of the clusters from the analysis?
- Similarly was there any way to stratify the clusters by prevalence to expand upon the suggestion that the high transmission setting limits the ability to detect a different serological profile?

Minor Comments

- Seropositivity to Pf was determined as being positive to any antigen, yet MSP-1 and CSP are likely to reflect very different exposure windows, with CSP being more recent. Did you look at seropositivity by stratifying for short / long-term markers of exposure?
- Considering the challenges of applying serology in high transmission settings, do the authors have any suggestions for prevalence ranges where it may be more appropriate?
- Transient reduction in Pf IgG seroprevalence looks interesting. Where you able to stratify the areas based on incidence e.g. health facility, to see if there was an association between incidence and probability of seroconverting?
- Are there any other plausible mechanisms (infectious diseases) that were not examined in this study that could have an azithromycin-mediated effect?
- While not significant, you identified larger changes in seropositivity between the arms for the longitudinal samples than for the general population. Could perhaps expand on when this would be appropriate to pursue this approach in the discussion.

RESPONSES TO REVIEWER COMMENTS

Reviewer comments are in **bold**, responses in normal font, and quoted text from the manuscript is in *italics*. Line numbers reference the clean (untracked) version of the manuscript.

Reviewer #2 (Remarks to the Author):

In this study the authors performed a follow-up analysis for the MORDOR trial in Niger, which found that mass distribution of azithromycin to preschool aged children reduced all-cause mortality by 18%. Their objective was to examine potential mechanisms for mortality reduction. They did this by estimating differences in IgG antibody responses to eleven pathogens between azithromycin and placebo groups 6 months after each round of treatment. They found high levels of background transmission and, despite high levels of heterogeneity in IgG responses, they found a small but significant reduction in *Campylobacter* spp. force of infection in the communities that received azithromycin.

This manuscript is well written, the analyses are thoughtfully laid out and implemented, and the presented results support the authors' conclusion that reduction in *Campylobacter* infection may be one of the reasons for reduction in mortality following azithromycin treatment. The two main things that were missing for me in the paper were: (1) potential sources of heterogeneity in the IgG assays and responses, and their implications for inferring a mechanism, and (2) what can be done with this new information going forward. I have described these and other comments in detail below.

Response: Thank you very much for your careful assessment of the study and your constructive suggestions for improvement.

Main comments:

1) It's a bit unclear to me whether the signal for *Campylobacter*, as oppose to other pathogens, is due to something biological/epidemiological and/or whether it's a reflection of the assay sensitivity or timing to detect this particular pathogen. The independent metagenomic study supports their conclusions, but I'm wondering if this is something the authors can also interrogate in their serological data. Is there information on how sensitivity to detecting these infections varies or on duration of the bacterial and protozoal antibody responses? Were there differences in between- or within-plate variation for the different antigens in the multiplex bead assay?

Response: IgG responses are generally a sensitive marker of previous infection for the pathogens included in this analysis. There were not large differences in between- or within-plate variability

for the different antigens in the multiplex assay — all had low CV%s (details below reported in the main text in response to **Reviewer 2, comment #4**).

We suspect the main limitation of IgG assays in this context would be their lower sensitivity to measuring short-term impacts of azithromycin on infection, which has been our focus in the Discussion. Furthermore, we suspect that intervention studies in high transmission settings could miss modest intervention effects due to IgG boosting through repeated infections, even if therapeutic interventions such as azithromycin transiently reduce IgG or prevent secondary infections through more rapid cure of the primary case.

We therefore expect that there were reductions in IgG responses to *Campylobacter*, but not other pathogens, either (a) because the intervention had a sufficiently large effect on *Campylobacter* infection and transmission that could still be detected through infrequent IgG measurements or (b) azithromycin treatment did not sufficiently impact IgG responses to other pathogens, even if it potentially had modest effects that may have been clinically relevant to child survival.

In response to this comment, and to several reviewer comments below that touched in on these same themes, we made the following additions to the analyses and manuscript text:

First, we completed a new analysis that separated the *P. falciparum* IgG responses into antigens thought to produce a more durable IgG response (MSP-1, AMA1) and those that produce a less durable response (GLURP-Ro, LSA1, CSP, HRP2). The analysis is summarized in new SI Figure 7 and found modest effects of azithromycin in both subgroups of antigens, suggesting that for *P. falciparum* IgG half-life is probably not the driving factor masking intervention effects. Response to **Reviewer 4, comments #2 and #6** includes text additions.

Second, we completed an exploratory subgroup analysis (clearly labeled as such) that stratified communities by baseline malaria parasitemia, grouping 17 communities with parasitemia $\leq 5\%$ and 13 communities with parasitemia $> 5\%$ — a cutoff chosen without regard to outcomes but only with regard to the natural break in the distribution of baseline parasitemia and the roughly equal split in communities between the two groups. This stratified analysis showed that there was a reasonably large, borderline insignificant difference of azithromycin on *P. falciparum* seroprevalence and force of infection in lower transmission communities with no effect in higher transmission communities. We summarized results in the new SI Figure 8. These results are consistent with the possibility that IgG boosting through repeated infections could mask intervention effects given the infrequent measurements in the trial (with boosting less likely in lower transmission communities, and thus differences in IgG more apparent). Response to **Reviewer 5, comment #2** includes text additions.

Third, we added more clarification in the Discussion's limitations regarding the interplay of transient effects from azithromycin, IgG half-life and IgG boosting with respect to the specific half-life of enteric pathogens. Also see our response to **Reviewer 4, comment #2**.

Discussion, Lines 252-269 , additions underlined

*“A second limitation was that the study collected blood spots approximately 6 months after each treatment administration. If azithromycin had short term effects on infection and antibody response, without effects on longer term pathogen carriage, then this study could have missed such transient effects. The timing of monitoring visits was chosen to provide longer-term data about community antibiotic resistance (i.e., resistance that persisted even months after the mass drug administration).⁶ However, azithromycin would be expected to have its strongest effect within days or weeks of administration. An analysis of timing of mortality in MORDOR Niger suggested that most of the intervention effect occurred within three months of distribution.³⁴ Among children ages 0-24 months, macrolides have been shown to reduce the risk of *Campylobacter* detection in stool significantly 0-15 days after treatment with reduced protection through 45 days, and no protection thereafter.²⁸ An analysis of enteric pathogen antibody dynamics among children suggested IgG half-life estimates in the range of 10 weeks, implying time to seroreversion of approximately one year absent repeated infections.⁸ Thus, the trial could have missed transient effects of azithromycin on pathogen infection, even if clinically significant with respect to child survival, without sustained reductions in community-level transmission because of slow waning of IgG responses following infection or IgG boosting after new infections between visits. Longitudinal, molecular measures of infection immediately before and within 1-2 weeks of treatment might provide a more definitive measure of azithromycin’s effect on pathogen infection.”*

Similarly, what are other potential sources of IgG variation between participants? The authors touch on several limitations of their serological approach and the timing of the data collection, but I’m wondering if some additional sources of variation could be examined (and if not here, then in a future study). In particular, differences in malnutrition, whether or not children actually received treatment, access to other health services, and other co-infections or co-morbidities, all of which could impact antibody responses as well as child mortality.

Response: We agree that there could be other sources of variation in IgG response between children beyond age and infection. The present trial accounted for age in the seroconversion rate modeling estimates, but as the reviewer notes there are probably other important determinants of variation in a child’s IgG response. We felt that the paper has already included a very large set of analyses focused on the RCT treatment effects. Since the comparisons are based on random assignment of azithromycin between communities, and communities were well balanced based on measurable characteristics, any other sources of variation would not likely cause bias. Instead, they would be useful exploratory, observational analyses appropriate for future analyses. We

have made all data associated with this part of the trial public to help facilitate future analyses such as those suggested in this comment. Source data are available through OSF and Dryad: OSF: <https://osf.io/954bt/> Dryad (curation in progress): <https://doi.org/10.7272/Q6VX0DSD>

2) The *Campylobacter* finding confirms previous studies and suggests a plausible mechanism for reduction in mortality, but I'm not sure how this information can be used to inform treatment or other interventions. If the authors have additional ideas, it would be helpful to include them. If further study is needed, then it would be helpful to know more specifically what those next steps should be.

Response: Unfortunately, the primary prevention measures for *Campylobacter* infections have proven very difficult to implement to a level that actually reduces infection (members of our team have worked extensively in WASH interventions that have yet to succeed). Targeted antibiotics treatment, with potentially lower possibility for antimicrobial resistance selection, is another potential alternative. To our knowledge, there are no large-scale efforts focused on a vaccine. We added the following focused on primary prevention of *Campylobacter*:

Lines 214-217

*"This result motivates further study of optimal treatment and primary prevention measures to reduce *Campylobacter* transmission, which is a persistent challenge. For example, intensive intervention trials of improved household water, sanitation and handwashing conditions have failed to reduce infection among children."³¹⁻³³*

Another implication of their results, which the authors include in the discussion, is that for multiplex antibody assays to be useful as trial endpoints, they should be done using detailed longitudinal measurements. This is concrete information that can be used in future studies.

Response: We agree. No changes made in response, as we devoted already a fair amount of space in the Discussion to this topic.

Additional comments:

3) Why was this particular multiplex IgG panel selected? Was it because azithromycin has antimicrobial activity against each of these pathogens? If so, it would be helpful to connect those dots in the introduction.

Response: The panel was selected as the intersection between a larger library of antigens that members of our team at the US CDC had developed for integrated serologic surveillance and

those that we felt could plausibly be impacted by mass azithromycin distribution. We added this clarification to the Introduction:

Lines 75-77

“The IgG panel was selected from a larger library of possible antigens that members of our team had previously developed for integrated serologic surveillance in low resource settings and could serve as plausible endpoints for mass azithromycin treatment.”⁷⁻¹⁰”

4) Buffer-only background absorbance levels were subtracted from MFI measurements in the multiplex bead assay. Were any other positive or negative controls or dilution series included on each plate that could be used to measure (and potentially adjust for) plate-to-plate variation?

Response: In the revision, we provided more details about the controls we used and have reported CV% across antigens used in the study. We added the following to the Methods:

Lines 369-379

“DBS samples were masked and randomly ordered by the UCSF trial coordinating center before sending them to the laboratory at the CDC. Samples were run on 63 plates and each plate included positive controls from a high positive sera pool (1:400 dilution), a low positive sera pool (1:6400 dilution) and a normal human sera (single) control. To assess plate-to-plate variation, we estimated the standard deviation (SD) for each antigen’s responses across plates. The laboratory protocol specified that a plate should be re-analyzed if more than half of the antigens had SDs that deviated by >20% from the overall average in multiple controls. Two of 63 plates failed this criterion for one control but passed in the other two controls and were thus not repeated. For the positive control sample responses to the 22 antigens used in this study, the average CV% was 10.0 with a standard deviation of 4.9. The median CV% was 9.3 with a range of 2.9% to 20.2%.”

5) What were the panels of known seronegative samples for malaria, Cryptosporidium, and Giardia? Were these from this population in Niger or elsewhere? It would be helpful to have more details on this part of the methods. I’m also curious, would moving the seropositivity cutoffs up or down ~20% impact the overall conclusions?

Response: For malaria, presumed seronegative samples were obtained from anonymous blood donors in North America. For *Giardia*: ROC curves were generated using a panel of sera from North American giardiasis cases and negative controls (total N = 32). For *Cryptosporidium*: ROC curves were generated from a panel of North American sera previously shown to be positive or negative using the ‘gold standard’ large-format Western blot (total N = 68). We clarified the source of the specimens and included references in the methods.

Lines 394-399

“We converted IgG responses to seropositive and seronegative classes using seropositivity cutoffs derived from the mean plus 3 standard deviations (SD) of responses from a panel of 92 anonymous, adult, USA resident blood donors (all malaria antigens), from ROC-derived cutoffs based on responses from known positive and negative specimens from North America (Cryptosporidium n = 68, Giardia n = 32),³⁹ or from the mean plus 3 SD of presumed unexposed measurements (all other antigens).”

In response to the reviewer’s suggestion, we conducted a sensitivity analysis that varied seropositivity cutoffs +/- 20% and re-estimated the difference in seroprevalence between groups and the hazard ratio. We have provided this additional analysis as new SI Figure 14. Adjusting the seropositivity cutoffs +/- 20% led to small changes in level of seroprevalence for each pathogen, but did not qualitatively change our findings. We added the following description of the sensitivity analysis to the Methods and Results:

Methods Lines 502-503

“We assessed the sensitivity of the results to the seropositivity cutoff for each antigen by changing cutoff values +/- 20% and then repeating the primary analyses.”

Results Lines 179-180

“Adjusting seropositivity cutoffs by +/- 20% did not qualitatively change the results (SI Figure 14).”

Reviewer #4 (Remarks to the Author):

The work presented in this manuscript is of high significance for infectious disease intervention trials and the application of serological endpoints to aid in study evaluation. The following comments indicate areas where more clarity or details could be provided by the authors.

Response: We appreciate the reviewer’s thorough assessment and constructive comments.

Results

Study population and setting

1. pg 4 “antibody substudy enrolled 3,814 children...5,642 blood specimens.”

o While it is indicated in the caption of Figure 1 that there is a random sample of 40 children from each of the 30 communities in the trial, it would be good to note this here also.

o Additionally, would be good to confirm here that the random sample for the sub-study was representative of the same age distribution as the main study.

Response: We added this information to the text:

Lines 90-93

“The antibody substudy enrolled 3,814 children aged 1-59 months and tested a total of 5,642 blood specimens through the 36-month follow-up between March 2015 and June 2018 (Figure 1). Enrolled children comprised a random sample of up to 40 from each community at each visit and had the same age-structure as the overall trial.”

2. pg 4 “specimens were collect March – July toward the end of the dry season”

o It would be important to note potential differences in antibody decay rates both between pathogens and between antigens for the same pathogens. It is mentioned briefly in the Discussion, but this could have a large bearing of future selection of markers that are best used for evaluation of intervention effects over short period of time.

Response: This idea was also mentioned by **Reviewer #2 comment 1** and is reinforced below with **Reviewer 4 comment #6**. We addressed it through two additions to the paper. First, we clarified how *P. falciparum* antigens broadly group into those thought to elicit more- and less-durable responses. We added an analysis that separately evaluated *P. falciparum* seroprevalence and seroconversion rates for more- and less-durable markers (new SI Figure 7). We found that more durable responses dominate the overall epidemiologic patterns, but that the relative hazard between groups was similar for the two antigen groups (see also our response to comment #6, below). Second, we added details about estimates of enteric pathogen half-life to the Discussion limitations and more clearly explained why IgG responses that are relatively short lived might still be imperfect in a setting of assessing therapeutic treatments. We also suggest that molecular measures of infection, timed within 1-2 weeks of treatment, would likely provide a more definitive measure of effects.

Addition to the Discussion, Lines 262-269

“An analysis of enteric pathogen antibody dynamics among children suggested IgG half-life estimates in the range of 10 weeks, implying time to seroreversion of approximately one year absent repeated infections.⁸ Thus, the trial could have missed transient effects of azithromycin on pathogen infection, even if clinically significant with respect to child survival, without sustained reductions in community-level transmission because of slow waning of IgG responses following infection or IgG boosting after new infections between visits. Longitudinal, molecular measures of infection immediately before and within 1-2 weeks of treatment might provide a more definitive measure of azithromycin’s effect on pathogen infection.”

**3. pg 4 “antibody responses to ETEC LTB and cholera CTB were highly cross-reactive”
o Pf and Pv MSP antigens have also been known to have some degree of cross-reactivity?
Confirmation that this was not the case for those used in this study would be helpful**

Response: In response to this comment, we expanded SI Figure 4 so that it displays all pairwise correlations in antibody responses for the full multiplex panel. The augmented figure shows that there was a considerable amount of correlation between different *Plasmodium* spp. antibody responses (range of correlation 0.30 to 0.53). However, we note that correlation in antibody responses is necessary but not sufficient to identify cross-reactivity between antibody responses because children who are more likely to be infected with *P. falciparum* are also more likely to be infected by other malaria species through shared environmental and behavioral risk factors. We updated the Results to include this additional information:

Results, Lines 111-113

*“Moderate correlation between malarial antibody responses suggests that *P. vivax* or *P. ovale* responses might reflect some limited cross-reactivity from *P. falciparum* infections (SI Figure 4) However, co-infection with multiple malaria species in this highly endemic region cannot be ruled out.”*

**4. pg 5 “evidence of very low exposure to P.vivax or P.ovale”
o Related to the issues of Pf/Pv cross-reactivity, the low prevalence here may help indicate lack of substantial cross-reactivity, but on the other hand, sero-prevalence for Pv may be artificially low depending on the negative population used to select seropositivity thresholds and if the negative population used differs between the Pf and Pv antigens. See comments on Methods section on cut-off method clarifications.**

Response: As shown in the revised SI Figure 4 (and per the last comment, **Reviewer 4 comment #3**) there was some correlation in IgG responses across *Plasmodium* spp. antigens. Adjusting the seropositivity cutoff -20% only increased *P. vivax* seroprevalence from 1.3% to 1.5% (an increase of 9 individuals classified as seropositive) so we suspect that the specific cutoff choice did not have a substantial influence on *P. vivax* seropositivity estimates. In the revision we clarified the details of the malaria negative population used to derive cutoffs — we used the same 92 anonymous samples (North American adults) for seropositivity cutoff derivation across the *Plasmodium* spp. antigens.

Lines 394-399

“We converted IgG responses to seropositive and seronegative classes using seropositivity cutoffs derived from the mean plus 3 standard deviations (SD) of responses from a panel of 92 anonymous, adult, USA resident blood donors (all malaria antigens), from ROC-derived cutoffs based on responses from known positive and negative specimens from North America

(*Cryptosporidium* $n = 68$, *Giardia* $n = 32$),³⁹ or from the mean plus 3 SD of presumed unexposed measurements (all other antigens).

See also the new sensitivity analysis in response to **Reviewer 2 comment #5** (above) that varied the cutoffs by +/- 20% for all primary analyses, which showed results were robust to cutoff (SI Figure 14).

5. pg 5 “heterogeneity between study communities in the longer-lived MSP1.19 and AMA1 antibodies, and overall seroprevalence go shorter-lived Pf antibodies”

o This sentence could be reworded so it is clear whether the heterogeneity is referring to variation in village/community level sero-prevalence to the same antigen, or variation in sero-prevalence to different antigens. Or both?

Response: We clarified that heterogeneity refers to between-community heterogeneity in seroprevalence. The revised text is now:

Lines 114-116

*“There was heterogeneity between study communities in seroprevalence to longer-lived MSP-119 and AMA1 antibodies, and overall seroprevalence to shorter-lived *P. falciparum* antibodies (GLURP-R0, LSA1, CSP, HRP2) was considerably lower.”*

o It would be helpful to cite literature on which MSP/AMA are considered longer-lived and GLURP, LSA, HRP are considered shorter-lived

o A mention that HRP2 is not a Pf antigen, but a by-product of parasite biomass would be good here. It would be useful to mention, if this is measuring Ab responses to circulating HRP in bloodstream, and whether these Ab tend to be longer or shorter-lived than Ab responses to parasite-specific antigens.

Response: We have added a reference to a recent, quantitative review article that summarized IgG responses to all of the antigens considered in this panel (van den Hoogen et al. 2020). We clarified some of the antigen differences in the methods

Introduction Lines 75-79

*“The IgG panel was selected from a larger library of possible antigens that members of our team had previously developed for integrated serologic surveillance in low resource settings and could serve as plausible endpoints for mass azithromycin treatment.⁷⁻¹⁰ The *P. falciparum*, panel included a mix of antigens known to induce both long-lived (MSP-1, AMA1) and short-lived (GLURP-R0, LSA1, CSP, HRP2) IgG responses.¹¹”*

Methods Lines 347-350

“The panel included malaria antigens to various stages of P. falciparum infection, including sporozoite (CSP), hepatocyte (LSA1), merozoite (GLURP-R0) and erythrocyte (MSP-1₁₉, AMA1, HRP2), with MSP-1₁₉ and AMA1 thought to induce longer-lived IgG responses compared with other included P. falciparum antigens.”¹¹”

6. pg 5 “Antigen-specific, age-seroprevalence curves showed similar overall patterns between groups, with largest reductions in P. falciparum AMA1”

o In supplementary figure SI Figure 1, 4 out of the 6 Pf antigens show an average MFI across all ages that is below the seropositivity threshold, other than MSP1.19 and AMA1. Hence, is the overall sero-prevalence result here almost entirely due to MSP and AMA? What added information do the other antigens provide, if any?

Response: In response to this query, we conducted an additional analysis of *P. falciparum* responses, creating composite seropositivity indicators using IgG responses thought to be more durable (MSP-1, AMA1) and those less durable (GLRUP-Ro, LSA1, CSP, HRP2). The more durable responses had higher overall seroprevalence, as expected, but both groups of antigens exhibited reduced seroprevalence and lower seroconversion rates in the azithromycin group compared with placebo (albeit not statistically significant at the 95% level). We summarized this new, exploratory analysis in new SI Figure 7.

We made the following additions to the text:

Results, lines 128-131

“An exploratory analysis that grouped antigens into more durable (MSP-1, AMA1) and less durable (GLRUP-Ro, LSA1, CSP, HRP2) responses showed a slightly larger shift in age-seroprevalence curves for longer-lived responses but the relative reduction in seroconversion rate was similar across both sets of antigens (SI Figure 7).”

Methods, lines 404-406

“An exploratory analysis (not pre-specified) grouped P. falciparum responses by antigens with more durable IgG responses (MSP-1, AMA1) and less durable IgG responses (GLRUP-Ro, LSA1, CSP, HRP2).”

Effect on bacterial and protozoan antibody response

7. pg 5 “Campylobacter and ETEC seroprevalence were >90% among children 6-24 months, with little heterogeneity between communities

o Same comment as 5 for malaria above on clarifying what heterogeneity is referred to.

Response: We clarified that heterogeneity referred to between-community heterogeneity in seroprevalence. The revised text is now:

Lines 140-141

“Campylobacter and ETEC seroprevalence were >90% among children 6-24 months, with little heterogeneity in seroprevalence between communities (Figure 3a).”

Additional analyses

8. pg 6 “Children included in multiple cross-sectional samples between ages 12-59 months contributed to longitudinal analyses of malaria seroconversion (919 children, 2,197 measurements). Longitudinal samples from children ages 6-59 months (1,038 children, 2,516 measurements) contributed to analyses of Salmonella and Streptococcus, and longitudinal samples among children 6-24 months (313 children, 680 measurements) contributed to analyses of the remaining enteric pathogens.”

o What was the median (and range) of measurements per child in each set of longitudinal analyses?

Response: We added this additional information to the text, in the Methods section per **Reviewer 4, comment #14** (below)

Lines 469-475

“Children included in multiple cross-sectional samples between ages 12-59 months contributed to longitudinal analyses of malaria (919 children, 2,197 measurements, median [range] measurements per child 2 [2, 5]). Longitudinal samples from children ages 6-59 months (1,038 children, 2,516 measurements, median [range] measurements per child 2 [2, 5]) contributed to analyses of Salmonella and Streptococcus, and longitudinal samples among children 6-24 months (313 children, 680 measurements, median [range] measurements per child 2 [2, 4]) contributed to analyses of the remaining enteric pathogens.”

9. pg 6 “most comparisons were slightly underpowered given the small size of the longitudinal cohort”

o Was there a method used here to estimate study power / sample size for this analysis? If not, would be useful to include in the Discussion the challenges in determining correct sampling frame to achieve study power in these type of analyses.

Response: When designing the study, we did not estimate detectable effects for the longitudinal analyses since they were additional analyses. We inferred that the study was slightly underpowered based on the 95% CIs. In principle, it would not be more difficult to do sample size and power calculations for an incidence rate ratio comparison, using prospective seroconversion rates and appropriate measures of variability, such as described in Hayes & Moulton 2017, chapter 7.

Hayes, R. J. & Moulton, L. H. *Cluster randomised trials*. (Chapman and Hall/CRC, 2017).

Discussion

10. pg 7 “azithromycin had limited effects on antibody-based measures of pathogen exposure against a backdrop of very high transmission for most pathogens.”

o Is this implying that antibody responses and sero-prevalence have saturated for these pathogens in these transmission intensities? Would this be affected by methods used to determine sero-positivity cut-offs, e.g. selection of negative population?

Response: Yes, we expect much of the antibody signal was saturated in this very high transmission setting — seroprevalence to most pathogens we studied was extremely high in this study, and so would require a very strong intervention to reduce it enough to see differences between groups. We do not believe the saturation was a result of choice of seropositivity cutoffs, as the patterns in IgG levels (MFI-bg) showed very similar patterns and results in between-group comparisons (SI Figures 5, 9), and the new sensitivity analysis that varied cutoff levels by +/- 20% also found very consistent results (see **Reviewer 2, comment #5**, new SI Figure 14).

11. pg 8 “Previous results from this trial found that azithromycin reduced mortality attributed to malaria, diarrhea, dysentery, and pneumonia [3], and reduced malaria parasitemia”

o It may be useful to include these non-serological results in supplement as a direct comparison against the IgG data, or to show the degree of correlation between the serological vs. molecular of parasite measures

Response: We included a new supplemental figure (SI Figure 13) that summarizes community level *P. falciparum* seroprevalence by malaria parasitemia (thick smear microscopy). There was modest correlation between measurements (Spearman’s $\rho = 0.45$). We included the following additions to the results and methods in the text:

Results, Lines 178-179

“At the community level, seroprevalence was correlated with thick smear malaria parasitemia (Spearman’s $\rho=0.45$, SI Figure 13).”

Methods, Lines 382-386

“At the time of dried blood spot collection, a drop of blood was used to prepare a thick smear slide that was stained with 3% Giemsa and later assessed for the presence and density malaria parasites as previously described.⁴ We estimated community level malaria parasitemia as the proportion of children with positive thick smears by microscopy (consensus across three independent, masked assessors).”

Methods, Lines 423-424

“We compared community level malaria parasitemia prevalence and seroprevalence to P. falciparum antigens using Spearman rank correlation.”

12. pg 8-9 “our finding of no clear reduction in IgG responses to malaria and bacterial pathogens beyond Campylobacter suggests that in high transmission settings, infrequent IgG measurements by themselves are potentially an insensitive trial endpoint.”

o It may likely be the latter – if antigens used have antibody decay rates that are much longer than the time period of evaluation, or do not decay rapidly enough to detect change in an intervention trial. In the case of long-lived Ab responses (or rather Ab responses that maintain a stable MFI for a long period as opposed to Ab responses that wane rapidly but still remain at a low level), more frequent measures may be unlikely to be more useful as a trial endpoint. What is needed are antibody responses that boost and decay rapidly during a period of time that matches trial follow-up.

Response: We agree with the reviewer’s assessment. In the revision, we more directly discussed the possible use-case for IgG-based endpoints in interventions that prevent incident infections and overall community-level transmission (likely useful) versus interventions that treat existing infections (potentially less useful).

Lines 232-245

“Even if azithromycin successfully treated infections from these pathogens, IgG responses might be an insensitive marker of efficacy if treatment did not reduce the IgG response. For example, IgG longevity or low-density infections that elicited a robust adaptive immune response could explain the small reductions observed in P. falciparum IgG responses despite lower levels of parasitemia and parasite density in the azithromycin group.⁴ Longitudinal IgG measures among infected children who receive treatment would be needed to definitively assess whether IgG levels drop following treatment. IgG responses to Chlamydia trachomatis dropped only slightly after azithromycin treatment among children in Tanzania,³⁴ but similar studies have not been done for the antigens included in this analysis. Exploratory analyses that stratified by baseline malaria parasitemia showed more consistent reductions in P. falciparum seroprevalence in lower transmission communities only (SI Figure 8). Based on these observations, we surmise that IgG responses might be more useful for measuring effects of interventions that prevent incident infections or interventions that treat infections in settings without frequent reinfection.”

13. pg 10 “If azithromycin had short-term effects on infection and antibody response, without effects on longer term pathogen carriage, then this study could have missed such transient effects.”

o It seems that rather than capturing transient effects, it would be more important to define the duration of protection offered by the interventions. Do the authors think there is

potential for longitudinal serological assessment (or at well-defined time points) be able to determine this better than non-serological measures? If so, what might be feasible study designs?

Response: Based on the results of this study, we suspect that longitudinal, molecular measures within 1-2 weeks of treatment would provide the most definitive measure of azithromycin effects on pathogen infection. Please see our response to the reviewer's previous **comment #12** regarding longitudinal IgG measures and their use to measure effects of different types of interventions. Below is an addition to the Discussion to address this point regarding molecular measures as a potential advantage in this particular type of intervention:

Lines 262-269

“An analysis of enteric pathogen antibody dynamics among children suggested IgG half-life estimates in the range of 10 weeks, implying time to seroreversion of approximately one year absent repeated infections.⁸ Thus, the trial could have missed transient effects of azithromycin on pathogen infection, even if clinically significant with respect to child survival, without sustained reductions in community-level transmission because of slow waning of IgG responses following infection or IgG boosting after new infections between visits. Longitudinal, molecular measures of infection immediately before and within 1-2 weeks of treatment might provide a more definitive measure of azithromycin's effect on pathogen infection.”

Methods

Study design and participants

14. pg 12 “Children who were randomly selected in multiple survey rounds contributed to longitudinal analyses.”

o This would be a good place to include details on the number of children who were sampled in multiple rounds, and the median/range of measurements per child.

Response: Thank you for this suggestion. Here, we provided these additional details.

Lines 469-475

“Children included in multiple cross-sectional samples between ages 12-59 months contributed to longitudinal analyses of malaria (919 children, 2,197 measurements, median [range] measurements per child 2 [2, 5]). Longitudinal samples from children ages 6-59 months (1,038 children, 2,516 measurements, median [range] measurements per child 2 [2, 5]) contributed to analyses of Salmonella and Streptococcus, and longitudinal samples among children 6-24 months (313 children, 680 measurements, median [range] measurements per child 2 [2, 4]) contributed to analyses of the remaining enteric pathogens.”

Seropositivity cutoffs

15. pg 15 “from a panel of known seronegative sera (malaria antigens).”

o How was sero-negativity determined? Were these non-endemic malaria naïve samples or samples from endemic areas? Authors should mention in the Discussion if this would cause any differences between sero-prevalence measures across different malaria markers. e.g., were there differences in negative populations used for Pf vs. Pv that led to very low sero-prevalence for Pv?

Response: In response to this comment and **Reviewer 2 comment #5** we added more details about the malaria panel used to define seropositivity cutoffs. We used the same set of anonymous donors in the USA for all malaria marker seropositivity cutoffs, so that would not explain the very low seroprevalence for Pv.

Lines 394-399

“We converted IgG responses to seropositive and seronegative classes using seropositivity cutoffs derived from the mean plus 3 standard deviations (SD) of responses from a panel of 92 anonymous, adult, USA resident blood donors (all malaria antigens)... ”

Age structured sero-prevalence and force of infection

16. pg 16 “The model included community-level random effects to allow for correlated outcomes within community.”

o Given that there are children with multiple samples in repeated cross-sectional surveys, was there a restriction on number of samples per individual in the sero-prevalence, FOI and hazard ratio analyses e.g., first sample per individual? Or if all samples per individual, accounting for non-independence/correlation by individual?

Response: We did not restrict the analysis to include only one observation per child, so in many cases a single cluster could have included multiple observations from the same child. In the analysis of FOI, we accounted for potential outcome correlation within-community by including a community-level random effect, which was a parsimonious way to account for clustering within a generalized additive model (GAM).

Since the only inference of interest is a fixed effect at the community level (azithromycin vs. placebo) and communities are the independent unit in the analysis, inclusion of a random effect for community should be sufficient to account for clustering within-community and should not bias the standard errors of the community level treatment effect (Schmidt-Catran and Fairbrother 2016).

Schmidt-Catran, A. W. & Fairbrother, M. The Random Effects in Multilevel Models: Getting Them Wrong and Getting Them Right. *Eur. Sociol. Rev.* **32**, 23–38 (2016).

In response to this suggestion, we attempted to repeat our FOI analyses using a multilevel mixed model, which included random effects for the community and a new random effect for child nested within-community. However, since there were > 3,000 individual children in the trial, attempting to fit a multilevel model with community- and child-level random effects led to models that failed to converge in nearly every case — both when using the `mgcv::gam()` R package (what we used in our primary analysis) as well as when using the `lme4::glmer()` R package. The reason models fail to converge is that the large number of children nested within community creates a very complex random effects structure. In one example where the model did converge with a community level-random effect and a child-level random effect (*P. falciparum* MSP-1), the hazard ratio estimates and 95% CIs were nearly identical:

P. falciparum MSP-1 FOI analysis, hazard ratio for azithromycin/placebo:

Community-level RE alone: HR (95% CI) = 0.91 (0.66, 1.25)

Community and child RE: HR (95% CI) = 0.90 (0.67, 1.20)

We therefore have left the analysis including only random effects for community, which should be appropriate (per the Schmidt-Catran 2016 reference, above). However, we have made note of this nuance and model limitation in the Methods:

Lines 457-465

“In some cases, the same child was measured multiple times during the trial and contributed multiple measures within a community over the course of follow-up (next section). We considered the inclusion of a child-level random effect in the age-structured seroprevalence model, nested within community, to model this additional potential source of outcome correlation. However, models that included this more complex random effects structure failed to converge in most cases. Since our target of inference was the hazard ratio for azithromycin-treated communities versus placebo, inclusion of a community-level random effect, the independent unit of analysis and inference, should lead to unbiased estimates of the standard error even without attempting to model additional sources of variation within the community.”⁴⁷

Subgroup analyses

17. pg 18 “We hypothesized that azithromycin could reduce antibody based measures of transmission more among younger children who were immunologically naïve and at later phases of the trial due to additional rounds of biannual MDA.”

o How was the hypothesized age threshold determined?

Response: Thank you for pointing out that we failed to carry forward the rationale for the age choice from the pre-specified analysis plan into the methods. This is the text from the pre-specified analysis plan:

From the SAP: A pre-specified subgroup analysis will examine the effect among children who were 6 months or younger at the start of the trial. IgG response tends to increase with age and transmission could be intense for many of the pathogens. Thus, it is possible that older children will already have high IgG levels to specific pathogens after multiple infections before the study that will not wane with treatment. In contrast, for children born since the start of the trial, treatment could reduce the severity or duration of infections, which in turn could result in a less robust IgG response. We define the subgroup as children 6 months or younger (including not yet born) at the start of the trial because age 6 months is approximately when we would expect children to begin to produce their own IgG response.

Newly added text to Methods

Lines 490-493

“ For the age-stratified subgroup, we focused on children 6 months or younger (including not yet born) at the start of the trial because age 6 months is approximately when we would expect children to begin to produce their own IgG response (additional details in the statistical analysis plan).⁴⁰”

Statistical power, detectable effects, and adjustments for multiple comparisons

18. pg 18 “assuming a sample of 15 communities per arm and 140 measurements per community over four rounds, seroprevalence of 65% (*P. falciparum*), a community-level ICC of 0.004, and a two-sided alpha of 5%, we estimated that we would have 80% power to detect a reduction of 5.4 percentage points in seroprevalence due to intervention.”

o Was the study powered specifically for Pf, and if so, what is the reason for this as opposed to other pathogens?

o Clarify that this is sero-prevalence to all Pf antigens combined?

o Was the ICC based on village/community-level measure of seroprevalence? If so, it would be useful to report the mean and SD here.

Response: As we noted in the manuscript, the trial was designed and powered around antimicrobial resistance monitoring. So, given the fixed trial design (30 communities, annual monitoring) we estimated the detectable effect for seroprevalence under a reasonable set of parameters. We used *P. falciparum* as an illustrative example since seroprevalence was closer to 50% and thus conservative compared with pathogens that had seroprevalence closer to 0% or 100%, and since we had an estimate of seroprevalence from a recent trial in Niger (PRET). We note in the main text that the analysis plan includes additional details regarding the assumptions and detectable effects.

This is the full text from the analysis plan:

We informed the sample size calculation with measurements from the PRET trial, where 991 children ages 1-5 years old were measured for antibody response to Plasmodium falciparum MSP-1 across 24 communities. In that study, mean seroprevalence to MSP-1 was 65% and the intra-class correlation coefficient for seropositivity was 0.004. We used a standard sample size equation for cluster randomized trials in binary outcomes (equation 7.11 of Hayes and Moulton).¹⁸

Under these assumptions, we estimated that with 15 communities per arm, 140 measurements per community (four phases) and 80% power, the minimum detectable relative reduction is 8% (prevalence difference = -5.4 percentage points). At 90% power, the detectable relative reduction is 10% (prevalence difference = -6.3 percentage points).

We estimated under the same assumptions that with 35 children per community (single phase), at 80% power the minimum detectable relative reduction is 14% (prevalence difference = -9.3 percentage points). At 90% power the detectable relative reduction is 16% (prevalence difference = -10.7 percentage points).

We clarified some of these points in the main text, additions underlined

Lines 509-515

*“The MORDOR morbidity monitoring trial was designed around the primary antimicrobial resistance monitoring endpoints.^{5,44} For the present analyses, assuming a sample of 15 communities per arm and 140 measurements per community over four rounds, seroprevalence of 65% (*P. falciparum* MSP-1), a community-level ICC of 0.004, and a two-sided alpha of 5%, we estimated that we would have 80% power to detect a reduction of 5.4 percentage points in seroprevalence due to intervention ⁴⁵ (the pre-analysis plan provides additional details, <https://osf.io/d9s4t/>, seroprevalence and ICC assumptions were estimated from the nearby PRET trial).⁵¹”*

Figures and Tables

19. Table 1. Baseline study group characteristics

- o Alongside overall sero prevalence by antigen, here would be a good place to also include community-level seroprevalence (mean, SD across all or by intervention)
- o N used for individuals in caption could be changed so it is not confused with N=communities in the columns of the table

Response: In Table 1, we moved the N used for individuals into a footnote so that it is not confused with the N=communities in the column headers. We created a new SI Table 3 that

summarizes community level means along with the SD in community-level seroprevalence and the intra-cluster correlation coefficient (ICC) for each seroprevalence endpoint. We added the following text:

Results Lines 176-178

“Intra-class correlation (ICC) estimates varied across antibodies, with higher between-community standard deviation and higher ICCs for longer lived malaria responses and Streptococcus group A (SI Table 3).”

Methods Lines 420-423

“We summarized the standard deviation of community-level seroprevalence and estimated the intra-class correlation coefficient (ICC) for community level responses using a mixed effects binomial model with a parametric bootstrap to estimate 95% confidence intervals for the ICC (1000 iterations).⁴¹”

Supplementary Material

20. SI Figure 3 – A. specimens collected by calendar month and study phase

o It is difficult to interpret this plot. Are there multiple phases in the same calendar month (column) due to variable enrolment date for individuals/villages?

Response: The caption for SI Figure 3 was confusing. We added some additional information to panel A’s caption to explain that the calendar months spanned multiple years — the purpose of the figure is to show alignment between surveys in the time of year and to show how the survey in phase 6 was the only set of measurements after the rainy season. The revised caption is now:

“SI Figure 3. Dried blood spot measurement timing and monthly rainfall in the MORDOR Niger trial. (a) Specimens collected by calendar month and study phase (months since baseline), showing that most blood spot specimens were collected March-July, except those in phase 6. (b) Monthly precipitation in study communities. Colored periods show the timing of dried blood spot specimen collection for each study phase. Created with notebook <https://osf.io/pem3z>.”

21. SI Figure 4 – Correlation between antibody responses for ETEC LTB and cholera CTB

o A similar plot for MSP between malaria species would also be useful to confirm there is limited cross-reactivity.

Response: Thank you for this suggestion. In response to this suggestion and the earlier suggestion (**Reviewer #4, comment 3**), we augmented SI Figure 4 to include the complete set of pairwise correlation estimates between all antibody responses measured in the multiplex.

Reviewer #5 (Remarks to the Author):

This study expands on the primary outcome from the MORDOR trial where a significant reduction in all-cause mortality was associated with biannual azithromycin treatment in under 5's, and attempts to identify the basis for this reduction through assessing changes in serological profiles between the two arms. The manuscript is well written, the experiments well performed and analysed. The conclusions are robust albeit limited in their ability to explain the overall impact of the trial. This may be due to the high transmission setting that could mask some of the studies impact, or be due to the study design (e.g. limited sampling).

Response: Thank you for this summary and for the constructive comments below.

Major Comments

1. Was any sensitivity analysis performed to assess the impact of removal of one or more of the clusters from the analysis?

Response: In response to this suggestion, we conducted a leave-one-out sensitivity analysis that re-estimated differences between groups in 30 subsets of the data, leaving out each community in turn. For each antibody response and in each subset of the data, we compared groups based on the difference in seroprevalence and the relative hazard of seroconversion, as in the primary analysis. We additionally calculated the jackknife estimate of the bias, which is $Bias = (n - 1)(\hat{\theta}(\cdot) - \hat{\theta})$, where n is the number of communities, $\hat{\theta}(\cdot)$ is the mean of the parameter estimate over the leave-one-out replicates, and $\hat{\theta}$ is the mean across all communities. The analysis is summarized in new SI Figure 15. The results show that the analyses were robust to exclusion of individual communities and that no single community had undue influence on the analysis. The jackknife estimates of the bias were very small, <0.03% for difference in seroprevalence and <0.03 in the log hazard ratio. We made the following additions to the Methods and Results:

Results Lines 180-182

“Leave-one-out sensitivity analyses showed overall estimates were unbiased and no single community had undue influence on the analysis (SI Figure 15).”

Methods Lines 502-506

“We assessed the sensitivity of the results to individual communities by conducting a leave-one-out analysis that examined the distribution of analysis results excluding each community in turn, and computed a jackknife bias estimate for the difference in seroprevalence and the log hazard ratio of seroconversion rates.⁴⁸”

2. Similarly was there any way to stratify the clusters by prevalence to expand upon the suggestion that the high transmission setting limits the ability to detect a different serological profile?

Response: In response to this suggestion, we added an exploratory subgroup analysis that stratified the *P. falciparum* serology analyses by community-level baseline malaria parasitemia. At baseline, communities had a range of parasitemia from 0% to 39% with 13 of 30 communities falling >5%. We stratified by above and below 5% baseline parasitemia since this was a natural break in the distribution and provided roughly balanced numbers of communities in the two strata (choice not made with regard to outcomes). In the stratified analysis, seroprevalence charted a more substantial effect of azithromycin among lower prevalence communities, and among higher prevalence communities there was no evidence for any effect, which was also reflected in very consistent shifts in age-seroprevalence curves. This does provide additional evidence to support the conjecture that IgG-based measures might be less valuable in very high transmission settings (at least with infrequent measurements). We summarized these results in a new SI Figure 8. We also included new text in the Results and Methods sections:

Results, Lines 132-137

“To assess whether serology might be a more sensitive endpoint in lower transmission settings, we conducted an exploratory subgroup analysis (suggested during peer review, not pre-specified) that stratified communities by baseline malaria parasitemia $\leq 5\%$ ($n=17$) versus $>5\%$ ($n=13$). There was modest evidence for larger reduction in IgG seroprevalence among communities with baseline parasitemia $\leq 5\%$ (difference = -11% , 95% CI -22% to 2%) with no difference between groups among communities with higher baseline parasitemia (SI Figure 8).”

Methods, Lines 496-499

*“An exploratory subgroup analysis stratified of *P. falciparum* IgG comparisons by baseline malaria parasitemia (not pre-specified). We estimated the proportion of children who were malaria positive by thick smear microscopy in the baseline survey and identified a natural break in the distribution with 17 communities $\leq 5\%$ parasitemia prevalence and 13 communities $>5\%$ parasitemia prevalence.”*

Minor Comments

3. Seropositivity to Pf was determined as being positive to any antigen, yet MSP-1 and CSP are likely to reflect very different exposure windows, with CSP being more recent. Did you look at seropositivity by stratifying for short / long-term markers of exposure?

Response: We did not originally plan for this analysis, but **Reviewer 4 comment #6** had this same question and suggestion. In response, we added an exploratory analysis that summarized Pf antibody responses for longer-lived responses (MSP-1, AMA1) and the shorter-lived responses

(all others). Results show that the longer-lived responses dominate the shape of the age-seroprevalence curve, but the estimated hazard ratio in the force of infection was similar to the two groups. We summarized this new, exploratory analysis in new SI Figure 7.

4. Considering the challenges of applying serology in high transmission settings, do the authors have any suggestions for prevalence ranges where it may be more appropriate?

Response: We could not interrogate the study for all pathogens across a range of transmission, but we did have individual-level measures of malaria parasitemia at baseline that we could use to assess this in the case of *P. falciparum*. In hypothesis generating, exploratory analyses suggested in reviewer 5 comment #2 (above), we found some evidence for larger impact on *P. falciparum* IgG outcomes in communities with lower baseline malaria parasitemia ($\leq 5\%$) and no effect in communities at higher baseline prevalence. We added the following text to the Discussion:

Lines 241-245

“Exploratory analyses that stratified by baseline malaria parasitemia showed more consistent reductions in P. falciparum seroprevalence in lower transmission communities only (SI Figure 8). Based on these observations, we surmise that IgG responses might be more useful for measuring effects of interventions that prevent incident infections or interventions that treat infections in settings without frequent reinfection.”

5. Transient reduction in Pf IgG seroprevalence looks interesting. Where you able to stratify the areas based on incidence e.g. health facility, to see if there was an association between incidence and probability of seroconverting?

Response: Yes. For details please see response to this reviewer’s earlier **comment #2** (above).

6. Are there any other plausible mechanisms (infectious diseases) that were not examined in this study that could have an azithromycin-mediated effect?

Response: It is plausible that azithromycin could reduce infection from clinically important enteric or respiratory bacterial pathogens that we did not measure in the antibody assay, for example *Shigella* or *Streptococcus pneumoniae*. Analysis of verbal autopsy records of deaths suggested reductions in pneumonia and meningitis, which could be caused by several species of bacteria not measured. We have been careful in our interpretation to be clear that the reductions in *Campylobacter* measured in this analysis support one possible mechanism for mortality reduction, but perhaps not all. We added this sentence to the Discussion:

Lines 218-219

“The multiplex assay included a diverse panel of antigens but did not cover all clinically relevant bacterial pathogens that contribute to child mortality, such as those that cause pneumonia.”

7. While not significant, you identified larger changes in seropositivity between the arms for the longitudinal samples than for the general population. Could perhaps expand on when this would be appropriate to pursue this approach in the discussion.

Response: For most outcomes, longitudinal and cross-sectional estimates were consistent, albeit the longitudinal-estimates were less precise given the much smaller sample. The figures below, which are included in the computational notebooks but not as a formal display in the paper, make this more clear.

analysis

- cross-sectional
- ▲ longitudinal

Reviewers' Comments:

Reviewer #2:

Remarks to the Author:

I thank the reviewers for their very thorough responses to the comments. My concerns have been addressed, specifically through the inclusion and discussion of SI Figures 7,8 and 14.

Reviewer #4:

Remarks to the Author:

Many thanks for the thorough edits and responses to the reviewer comments. The additional analyses was especially helpful in clarifying questions raised in the previous review.

In particular, consideration of the transmission intensities where serology may be most useful was interesting, and the potential effect of frequency of reinfection. As well as the sensitivity analysis of cut-offs on study outcomes.

All comments raised were suitably addressed.

A few minor comments, but they do not need to be edited necessarily to proceed to publication:

- The distinction between long and short-lived antibody responses may not be very clear. Previous studies, including the one cited, may sometimes consider GLURP as a marker of cumulative exposure alongside AMA and MSP, for example. CSP may often also have variable responses. However, for the purposes of this analysis, these distinctions are unlikely to have a large bearing on the final results.
- Would the sub-analyses of individuals <6 months be subject to additional IgG noise due to presence of maternal antibodies? It seems to contradict some of the previously stated methods of excluding individuals <12mo and <6mo for this same reason.

Reviewer #5:

Remarks to the Author:

The reviewers have satisfactorily responded to all of the comments I raised during the first round of reviews. The result is a more rounded and higher quality manuscript that is deserving of publication in Nature Communications.

RESPONSES TO REVIEWER COMMENTS

Reviewer comments are in **bold**, responses in normal font, and quoted text from the manuscript is in *italics*. Line numbers reference the clean (untracked) version of the manuscript.

Reviewer #2 (Remarks to the Author):

I thank the reviewers for their very thorough responses to the comments. My concerns have been addressed, specifically through the inclusion and discussion of SI Figures 7,8 and 14.

Response: No further changes.

Reviewer #4 (Remarks to the Author):

Many thanks for the thorough edits and responses to the reviewer comments. The additional analyses was especially helpful in clarifying questions raised in the previous review.

In particular, consideration of the transmission intensities where serology may be most useful was interesting, and the potential effect of frequency of reinfection. As well as the sensitivity analysis of cut-offs on study outcomes.

All comments raised were suitably addressed.

A few minor comments, but they do not need to be edited necessarily to proceed to publication:

- The distinction between long and short-lived antibody responses may not be very clear. Previous studies, including the one cited, may sometimes consider GLURP as a marker of cumulative exposure alongside AMA and MSP, for example. CSP may often also have variable responses. However, for the purposes of this analysis, these distinctions are unlikely to have a large bearing on the final results.

Response: We agree that potentially the whole GLURP protein, or other fragments, could be used as a long(er)-term marker of previous *P. falciparum* infection, but this study's data clearly show the Ro fragment has very low seroprevalence compared with AMA1 and MSP-1, consistent with shorter-lived IgG durability. We added the following reference to provide additional support that GLURP-Ro has been shown to have a short half-life in longitudinal studies of IgG decay (new reference #13): <https://pubmed.ncbi.nlm.nih.gov/33737926/>

We also agree that there may be some individual variation in CSP responses, which is true for all antibody responses in our experience. In previous studies by members of our team we have found CSP to provide very clear population-level summaries in areas of high transmission (e.g. <https://doi.org/10.1371/journal.pntd.0006278.g001>).

- Would the sub-analyses of individuals <6 months be subject to additional IgG noise due to presence of maternal antibodies? It seems to contradict some of the previously stated methods of excluding individuals <12mo and <6mo for this same reason.

Response: We clarified that the subgroup of children <6 months was based on the child age at the beginning of the trial, but that analyses only included IgG responses after children had aged into the windows that excluded maternal IgG contributions (<6 months for bacterial and protozoan pathogens, <12 months for malarial antigens). We made this clarification in the Supporting Figure 12 legend:

“Children who were younger than 6 months at the start of the trial may have not been born at the trial start but were enrolled in later visits. IgG responses were not included in the analysis until children were older than age 6 months (bacteria and protozoa) or 12 months (malaria) to avoid maternal IgG contributions (a pre-specified rule).”

Reviewer #5 (Remarks to the Author):

The reviewers have satisfactorily responded to all of the comments I raised during the first round of reviews. The result is a more rounded and higher quality manuscript that is deserving of publication in Nature Communications.

Response: No further changes.